# Guidance for good practice in the application of machine learning in development of toxicological quantitative structure-activity relationships (QSARs)

**Samuel J. Belfield**[ID][☯], **Mark T.D. Cronin**[☯], **Steven J. Enoch**[☯], **James W. Firman**[ID]*[☯]

School of Pharmacy and Biomolecular Sciences, Liverpool John Moores University, Liverpool, United Kingdom

☯ These authors contributed equally to this work.

* j.w.firman@ljmu.ac.uk

## Abstract

Recent years have seen a substantial growth in the adoption of machine learning approaches for the purposes of quantitative structure-activity relationship (QSAR) development. Such a trend has coincided with desire to see a shifting in the focus of methodology employed within chemical safety assessment: away from traditional reliance upon animal-intensive *in vivo* protocols, and towards increased application of *in silico* (or computational) predictive toxicology. With QSAR central amongst techniques applied in this area, the emergence of algorithms trained through machine learning with the objective of toxicity estimation has, quite naturally, arisen. On account of the pattern-recognition capabilities of the underlying methods, the statistical power of the ensuing models is potentially considerable– appropriate for the handling even of vast, heterogeneous datasets. However, such potency comes at a price: this manifesting as the general practical deficits observed with respect to the reproducibility, interpretability and generalisability of the resulting tools. Unsurprisingly, these elements have served to hinder broader uptake (most notably within a regulatory setting). Areas of uncertainty liable to accompany (and hence detract from applicability of) toxicological QSAR have previously been highlighted, accompanied by the forwarding of suggestions for "best practice" aimed at mitigation of their influence. However, the scope of such exercises has remained limited to "classical" QSAR–that conducted through use of linear regression and related techniques, with the adoption of comparatively few features or descriptors. Accordingly, the intention of this study has been to extend the remit of best practice guidance, so as to address concerns specific to employment of machine learning within the field. In doing so, the impact of strategies aimed at enhancing the transparency (feature importance, feature reduction), generalisability (cross-validation) and predictive power (hyperparameter optimisation) of algorithms, trained upon real toxicity data through six common learning approaches, is evaluated.

**Data Availability Statement:** All relevant data are within the manuscript, Supporting Information files and the public repostiory GitHub (https://github.

com/LJMU-Chemoinformatics/Best-Practice-Supplementary).

**Funding:** The author(s) received no specific funding for this work.

**Competing interests:** The authors have declared that no competing interests exist.

## 1. Introduction

Use of computational (*in silico*) approaches supporting the prediction of toxicological effect has become standard practice within modern chemical safety assessment. Quantitative structure-activity relationships (QSARs) are amongst the most well-established of the available *in silico* techniques and, as such, have been used extensively in order to identify potential hazard and predict potency [1]. QSAR models attempt to formalise the relationship between descriptors (quantities derived from the structural features and physico-chemical properties of molecules) and an endpoint property of interest [2]. Traditional QSAR was based predominantly upon regression analysis. However, as far back as the 1980s, a variety of other multivariate statistical approaches were being applied–with the uptake of neural networks following during the early 1990s [3,4]. The past decade has seen a greater shift towards employment of machine learning (ML) strategies in the development of predictive toxicology models [5]. There is no one reason for the increased use of ML: however, factors including heightened availability of data, more easily accessible informatics and statistics tools, and greater computational power have all contributed [6].

ML methods originated in the early to mid-20th century from mathematical considerations of data matrices, and have since developed primarily within the field of computer science. Emerging from pattern recognition studies and the concept of computational learning, ML algorithms can adapt and update during the process of training without explicit programming to do so–in turn improving predictive accuracy in an automated manner [7]. As such, they are now identified as one of the most vital and rapidly evolving areas in chemoinformatics [8,9]. Broadly, two overarching classes of ML may be distinguished: supervised or unsupervised. In this regard, the majority of QSAR applications adopt supervised learning approaches–whereby substances are labelled with the investigated property of interest (i.e., toxicological potency), and combinations of trends amongst a matrix of chemical descriptors are sought which might best relate to it. This stands in contrast to unsupervised techniques, through which patterns are identified from unlabelled data (generally useful in clustering exercises) [10,11]. Many distinct ML methodologies have been reported: some applicable solely to supervised tasks, some solely to unsupervised, and others to both. The major ML strategies relevant towards QSAR, as identified within the review of Lo et al., are outlined through Fig 1 (and further discussed in Section 2.3) [11]. Amongst these are the neural network (both deep and shallow), decision tree-derived ensemble learning (random forest and gradient-boosted trees), support vector machine and *k*-nearest neighbour approaches.

In ensuring the acceptability of a QSAR for use in prediction of toxicity, it is essential that model output be established as fit for a given purpose [12]. In practice, QSARs may be employed for tasks ranging from the rapid screening of large chemical libraries and inventories, to the identification of potential hazards relevant to risk assessment within individual substances (i.e., replacement for a specified test or else contribution to a weight of evidence evaluation). Legislation such as EU Regulation on Registration, Evaluation, Authorisation and restriction of CHemicals (REACH) states that a prediction should provide the same information as the test that it is replacing (the "adaptation of testing" requirement). To achieve this, amongst other criteria, the model must be demonstrated "scientifically valid". At present, assessment protocols such as the OECD Principles for the Validation of QSARs are applied in order to evaluate whether or not this is the case [13]. ML-based toxicity models can be challenging to appraise through these principles–a factor which has, unsurprisingly, led to reluctance amongst regulators to approve their broad acceptance. When related, for instance, to traditional regression analysis, they are commonly perceived to lack a defined and transparent algorithm (in violation of OECD Principle 2). Such opacity serves to obscure mechanistic

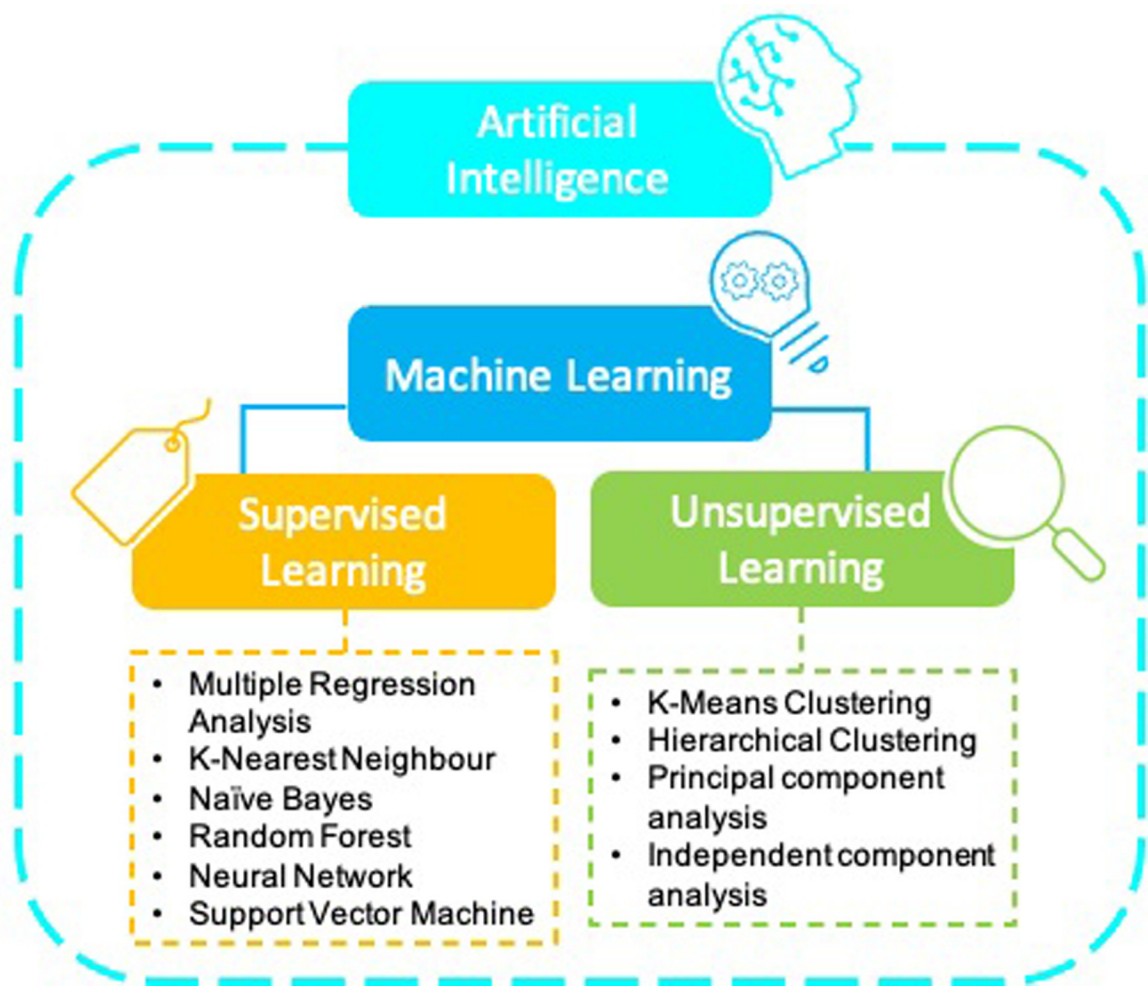

**Fig 1. Overview of machine learning approaches as generally applied within *in silico* toxicology.**

interpretability (OECD Principle 5)–whilst their general complexity may hinder provision of conclusive documentation (hence obstructing replicability) [14,15]. It is furthermore noted that generalisability of predictions may be called into question, on account of the tendency of the adopted techniques to overfit training data [16].

A 2022 review authored by Lin and Chou serves to highlight the potential versatility of ML as applied to toxicological modelling–outlining the construction of algorithms within a wide assortment of endpoints, each demonstrating a promising level of performance [17]. However, it is apparent that aforementioned challenges with respect to the reproducibility, ease-of-interpretation and capacity for generalisation in such methods must be appropriately addressed if their wider credibility (and by extension their uptake) is to be ensured. Recent work focusing upon the assessment of uncertainties associated with application of QSAR (itself drawing from the OECD QSAR Principles) could provide valuable insight into means through which this goal might, in practice, be realised [18]. Although the evaluation scheme devised by Cronin et al. for uncertainty appraisal is, in its present form, readily applicable towards a vast range of QSAR practices, additional supplemental guidance offering specific consideration of ML methodology will undoubtedly provide greater confidence in instances where such approaches are to be encountered.

The aim of this investigation was to identify aspects of best practice accompanying the development of ML methods for predictive toxicology application–an exercise facilitated through consideration of key areas of uncertainty both common and unique to them, as characterised within the Cronin et al. scheme [18]. To achieve this, two toxicity datasets of varying complexity and consistency were modelled using an assortment of standard ML approaches. The influence of state-of-the-art parameter optimisation and feature importance-detection techniques upon the predictive performance, generalisability and interpretability of the resulting algorithms was assessed–with focus placed upon ascertaining the benefits and potential shortcomings associated with adoption of each. In light of considerations arising from this exercise, extensions and amendments to Cronin et al. protocol were offered. It is intended that the following of the suggested principles should ultimately assist in improving the broader acceptability of ML within this field.

## 2. Materials and methods

Datasets and Python source code employed for and in the processes of model construction, optimisation and performance assessment are each freely accessible through the link: https://github.com/LJMU-Chemoinformatics/Best-Practice-Supplementary.

### 2.1. Sourcing and curation of toxicological data

Each of the two datasets adopted in this study were subject to preliminary curation, in order to ensure their suitability for use within the training of QSAR. Only discrete organic molecules with defined structure were eligible for inclusion: as such, inorganic substances, mixtures, polymers and those of ambiguous identity were excluded–as were duplicate entries. Salts and secondary fragments were, in addition, stripped. Structural information was expressed in the form of Simplified Molecular Input Line Entry System (SMILES) strings (www.daylight.com; [19]), canonicalised through use of OpenBabel software (v. 2.4.0; http://openbabel.org; [20]).

**2.1.1. *Tetrahymena pyriformis* growth inhibition dataset.** Data relating specifically to the acute toxicity of compounds towards the aquatic ciliated protozoan *Tetrahymena pyriformis* was extracted from the publication of Ruusmann and Maran [21]. In total, data describing 2,072 substances was retrieved–a number which reduced to 1,995 following performance of curation as outlined above. Explicitly, the endpoint examined was *T. pyriformis* population growth inhibition, expressed as the inverse logarithm on the millimolar concentration causing 50% inhibition in growth following 40 hours of treatment (for a general description of experimental protocol, please refer to work of Schultz) [22].

**2.1.2. Rat acute oral lethality dataset.** Data describing acute oral lethality towards the rat (LD50, expressed as $mmol/kg_{bw}$ and subsequently log-transformed), as presented in Gadaleta et al., were utilised [23]. This had been originally sourced by the National Toxicology Program (NTP) Interagency Centre for the Evaluation of Alternative Toxicological Methods (NICEATM) and United States Environmental Protection Agency (US EPA). Upon curation (again following aforementioned procedure), the number of substances was reduced from 8,488 to 8,186.

### 2.2. Calculation and selection of molecular descriptors

Physico-chemical and structural descriptors for the chemicals in both datasets were acquired using PaDEL software (v. 2.21; [24]). In total, 1,441 1D and 2D molecular descriptors were calculated. Uninformative or unsuitable (and hence redundant) descriptors were removed–including those containing missing values, or else exhibiting low variance (<0.01) as defined using *VarianceThreshold* from the *feature_selection* module within the

Python (v. 3.7.6; www.python.org) scikit-learn (v. 0.22.1; www.scikit-learn.org; [25]) library. Subsets of the original data were created through the exclusion of descriptors displaying excessive collinearity. To identify such, pairwise correlation coefficients were determined between each–with thresholds set either to 0.9, 0.8, 0.7, 0.6, 0.5, 0.4, or 0.3. The descriptor within the pair which reported weakest general correlation to the target quantity was omitted. Resultant subsets were granted identifiers in the form TH_XX (relating to *T. pyriformis*) or LD_XX (rat lethality), whereby "XX" denotes the exclusion threshold adopted: by way of illustration, TH_90 describes the subset of *T. pyriformis* data resulting from the application of threshold 0.9 (i.e., from the dropping of one feature within each pair displaying greater than 90% collinearity). Parent datasets are in turn referenced respectively as TH_Full and LD_Full. When training non-decision tree-based algorithms, feature values were standardised (through use of *StandardScaler*, from scikit-learn *preprocessing* module).

## 2.3. Training of machine learning models

This analysis saw the comparison of six established ML techniques applicable to QSAR. Through each of the approaches, regression models characterising both datasets were constructed within Python. Random forest, support vector machine and *k*-nearest neighbours were developed using scikit-learn, extreme gradient boosting with the package XGBoost (v. 1.2.1; https://xgboost.ai; [26]) and neural networks (shallow and deep) through the libraries Keras (v. 2.4.0; www.keras.io; [27]) and TensorFlow (v. 2.3.1; www.tensorflow.org; [28]). Shallow and deep neural networks were differentiated on account of the number of hidden layers incorporated into their architecture (one and two respectively). Both forms were nevertheless trained through use the stochastic optimiser Adam and the rectified linear unit (ReLU) activation function [29,30]. Each ML form examined is introduced briefly below (please refer to corresponding publications cited for more detailed perspectives).

**2.3.1. Random forest.** Random forest (RF) is an ensemble learning method which ascribes prediction based upon the outcomes of a collection of (typically several hundred) decision trees, each of which is constructed through application of bootstrap aggregation ("bagging") [31]. Respective trees are, as such, trained upon a random subset of samples, defined by a random selection of features–in turn ensuring that dissimilarity is likely to be present amongst them. Final outcome is determined by consideration of output from each: either through means of averaging (in instances of continuous data) or majority voting (categorical) [32]. The presence of many trees, all of which are derived in accordance with bagging, serves to greatly mitigate the risks of overfitting inherent within the classical, lone decision tree approach.

**2.3.2. Extreme gradient boosting.** Whilst RF rests upon concurrent generation of mutually-independent decision trees, the process of gradient boosting sees trees developed sequentially–each constructed with the primary intention of minimising residuals/misclassifications arising from its predecessor [33]. Such trees are typically shallower than those trained through the RF algorithm, and are thus considered a form of "weak learner". As an ensemble approach, ultimate prediction is (similar to RF) determined through the consensus of individual learner output (albeit weighted in accordance with its accuracy)–either by means of averaging or by voting. Extreme gradient boosting (XGB) represents an advancement over traditional gradient boosting techniques, incorporating modifications aimed at the minimisation of overfitting (for example, penalising excessive complexity) and the enhancement of scalability [26].

**2.3.3. Support vector machine.** The support vector machine (SVM) seeks to fit a hyperplane through *n*-dimensional feature space, serving either to best-fit continuous data, or else (when applied to categorical data) most efficiently separate between classes [34]. Integral to the

generation and alignment of the hyperplane across higher dimensions is the utilisation of kernel functions [35].

**2.3.4. *k*-Nearest neighbours.**   *k*-Nearest neighbours (*k*-NN) represents a comparatively simple distance-centred approach, whereby the properties of a test object are inferred from those of the *k* (a quantity freely definable by the model builder) training set members positioned closest to it within *n*-dimensional feature space [36]. Such distance is determined through application of an appropriate metric–of which several (including the Chebyshev, Cosine, Euclidean, Manhattan and Minkowski methods) find common usage [37]. In the handling of continuous data, a prediction is attributed to the target object through an averaging of the corresponding values held by its *k* neighbours. When categorising, the class dominant amongst these neighbours is ascribed.

**2.3.5. Neural network.**   The concept of the neural network (NN) draws inspiration from those neuronal frameworks forming key functional units within the central nervous systems of biological organisms. At the most fundamental level, actions of organic neurons are simulated through nodes: mathematical operators connected in series, each of which applies a non-linear "activation function" so as to transform inputs from its predecessors–subsequently generating a single output fed to successors [38]. Influence of one node upon another is dictated by the weighting of any connection present between the pair–a quantity optimised during the process of training, typically through application of a "back-propagation" algorithm.

A functioning NN consists of a sequence of layers, each of which is formed from a string of parallel nodes. Those within an input layer feed those constituting an intermediate "hidden layer", which in turn connects to a terminal output layer. No theoretical limit exists as regards the number of hidden layers which may be integrated into a setup: those NN incorporating only one may be identified as "shallow" (SNN), and those with greater than one as "deep" (DNN) [39]. The additional complexity intrinsic to the deep network lends itself, in principle, to the generation of the more sophisticated model [40,41].

## 2.4. Statistical performance metrics and evaluation of model predictivity

Quality of model performance was evaluated using the standard metrics $R^2$, MSE, RMSE and MAE.

**2.4.1. Predictivity against test data.**   In order to assess model performance against unseen data, cross-validation was employed [42]. As such, the impact of fold quantity, k (not to be confused with parameter *k* specific to *k*-NN), as relates to the apparent predictivity of models, was assessed. Algorithms were trained though use of each ML technique (adopting default, un-optimised hyperparameter sets) upon the *T. pyriformis* TH_90 data subset. Values of k ranging from two to 25 were analysed for their influence upon two fields: $R^2_{CV}$ (cross-validated $R^2$, representing average of the acquired $R^2$ figures relating to each fold during its hold-out) and inter-fold $R^2$ variability (disparity between minimum and maximum in fold-wise $R^2$). Interplay between each outcome was examined, in order that the figure of k offering optimal balance between both could be determined. As such, this k value was adopted in all modelling procedures. For details concerning average size of dataset fold (i.e., number of compounds contained) in accordance with k, please refer to S1 File.

**2.4.2. Estimation of overfitting liability.**   So that the extent to which a model was liable to overfit towards training data could be ascertained, the acquired $R^2_{CV}$ (representing predictivity upon test data) was subtracted from corresponding $R^2_{train}$ (quality of algorithm fit relative to training set)–with a larger residual value ($R^2_{over}$) indicating greater overfitting potential.

## 2.5. Hyperparameter optimisation for enhancement of model predictivity

Influence of the variation of hyperparameters (those distinct tuneable quantities governing algorithm learning processes) upon model predictive performance was examined in all adopted ML techniques. For this purpose, the *T. pyriformis* data subset TH_90 was employed. A list of evaluated parameters is presented within Table 1, together with outlines of the value ranges assessed. Definitions of each property may be accessed through official scikit-learn (https://scikit-learn.org/stable/supervised_learning.html; accessed 5[th] July 2022) and XGBoost (https://xgboost.readthedocs.io/en/stable/parameter.html; accessed 5th July 2022) package documentation.

The search for the combination of hyperparameter properties granting optimal performance was initially conducted manually. To facilitate this, each variable was adjusted in a stepwise manner over a pre-defined range–all others being held constant at default values (details of both are listed in Table 1). Resulting alteration in predictivity was noted, with those parameter quantities associated with highest $R^2_{CV}$ (at k = 10) taken to be preferred. Subsequently, these figures were combined into sets so as to produce the "manually optimised" forms corresponding to each respective modelling approach. Parameter variations associated with induction of significant predictivity drop-off during the aforementioned series of stepwise assessments were noted. Such knowledge was used to inform the boundaries of ranges examined within the automated protocols adopted following.

Two such approaches were employed: the randomised search algorithm (*RandomizedSearchCV* available through the *model_selection* module as present within scikit-learn) and the Bayesian optimisation software Optuna (v. 2.2.0; www.optuna.org; [43]). Whilst a traditional grid search sees rigorous testing of all possible alignments in permitted hyperparameter values in order to identify definitively the combination most favourable, a randomised search instead assesses only a selection, drawn at random, from amongst them (offering a reduced intensity in terms of computational load) [44]. Bayesian techniques again examine a subset of all possible arrays. However, these are not taken on an arbitrary basis–rather, preferred values are reached through an iterative process whereby prior and posterior performances actively influence the areas of parameter space explored [45]. Automated procedures were run over a series of 50 trials, with Optuna employing error minimisation for purposes of direction.

## 2.6. Feature Importance and model interpretability

In order to examine the contribution of individual descriptors towards influencing algorithm output, models were evaluated through each of two methodologies enabling inference of feature importance: permutation feature importance and SHapley Additive exPlanations. Models were themselves trained upon *T. pyriformis* subset TH_90 in accordance with procedures outlined in Section 2.3 –albeit with adoption of optimised hyperparameter sets identified through application of Optuna (a process described within Section 2.5).

**2.6.1. Permutation feature importance.**   Permutation feature importance (PFI) represents a model-agnostic approach, freely applicable to all ML techniques. In brief, importance is determined through evaluation of the decrease in model predictivity which results following random shuffling in the values of a sole feature [31,46]. As such, relationship between feature and output are separated, in a manner which offers insight into the (global-scale) significance of former towards latter. Assessment was performed through employment of the *permutation importance* function within scikit-learn.

**2.6.2. SHapley Additive exPlanations.**   SHapley Additive exPlanations (SHAP) represents an application of Shapley values (a concept originating from within the field of cooperative game theory), in order to provide definition of feature importance at the level of the individual

**Table 1. Information relating to hyperparameters applicable in each algorithm.** Title of parameter is listed, alongside the default quantities present within the adopted training software. Value ranges examined during processes of manual and automated optimisation (where appropriate) are listed–as are their preferred quantities, as identified through each tuning approach.

| Modelling approach | Hyperparameter | Default quantities | Quantity ranges examined | | Optimised quantities | | |
|---|---|---|---|---|---|---|---|
| | | | In manual optimisation | In automated optimisation | Manual | Automated | |
| | | | | | | Random search | Optuna |
| RF | max_depth | Automatic[b] | 1–50 | 10–30 | 15 | 30 | 27 |
| | n_estimators | 100 | 50–500 | 100–500 | 490 | 490 | 499 |
| | min_samples_split[a] | 2 | 2–20 | - | 3 | - | - |
| | min_samples_leaf[a] | 1 | 1–100 | - | 1 | - | - |
| | max_leaf_nodes[a] | Automatic[b] | 2–202 | - | Automatic | - | - |
| | max_samples[a] | Automatic[b] | 0.1–0.99 | - | 0.99 | - | - |
| SVM | Gamma | scale[d] | 0.0001–0.01 | 0.0012–0.003 | 0.00168 | 0.0012 | 0.00121 |
| | C[c] | 1 | 0.5–50 | 1–10 | 5 | 8.58 | 9.39 |
| | Epsilon | 0.1 | 0.001–1 | 0.001–0.02 | 0.418 | 0.018 | 0.00852 |
| k-NN | n_neighbors | 5 | 1–20 | 1–15 | 6 | 3 | 3 |
| | P | 2 | 1–5 | 1–3 | 1 | 1 | 1 |
| XGB | Eta | 0.3 | 0.005–0.5 | 0.1–0.15 | 0.107 | 0.1 | 0.103 |
| | min_child_weight | 1 | 1–20 | 1–10 | 7 | 4 | 2 |
| | max_depth | 6 | 1–50 | 2–8 | 4 | 4 | 5 |
| | Gamma | 0 | 0–3 | 0–0.3 | 0.103 | 0.1 | 0.00145 |
| | n_estimators | 100 | 50–500 | 100–250 | 250 | 250 | 205 |
| | Subsample | 1 | 0.1–1 | 0.8–1 | 1 | 0.8 | 0.816 |
| | colsample_bytree | 1 | 0.1–1 | 0.5–1 | 0.6 | 0.9 | 0.962 |
| | max_delta_step[a] | 0 | 0–10 | - | 0 | - | - |
| | lambda[a] | 1 | 0–1 | - | 0.778 | - | - |
| | alpha[a] | 0 | 0–10 | - | 3 | - | - |
| SNN[e] | Neurons | 512 | 50–1000 | 50–1000 | 400 | 550 | 601 |
| | dropout_rate | 0 | 0–0.5 | 0–0.5 | 0.1 | 0.2 | 0.444 |
| | Epochs | 100 | 50–500 | 50–500 | 100 | 250 | 236 |
| | batch_size[f] | 128 | 32–512 | 32–512 | 64 | 64 | 197 |
| | learn_rate | 0.001 | 0.0001–0.003 | 0.0001–0.001 | 0.001 | 0.0003 | 0.000376 |
| DNN[g] | neurons (hidden layer 1) | 512 | 50–1000 | 50–1000 | 750 | 650 | 944 |
| | neurons (hidden layer 2)[h] | 512 | 50–1000 | 50–1000 | 750 | 50 | 784 |
| | dropout_rate (hidden layer 1) | 0 | 0–0.5 | 0–0.5 | 0.2 | 0.3 | 0.161 |
| | dropout_rate (hidden layer 2)[h] | 0 | 0–0.5 | 0–0.5 | 0.2 | 0.4 | 0.494 |
| | Epochs | 100 | 50–500 | 50–500 | 100 | 500 | 498 |
| | batch_size[f] | 128 | 32–512 | 32–512 | 64 | 32 | 75 |
| | learn_rate | 0.001 | 0.0001–0.003 | 0.0001–0.001 | 0.001 | 0.0003 | 0.000321 |

a. Parameters not subject to automated optimisation.

b. Value of parameter defined by algorithm should the term "None" be entered (please refer to official scikit-learn documentation, linked within Section 2.5).

c. Within automated procedure, range 1–10 applicable to randomised search only (1–20 instead examined in Optuna).

d. Value of parameter defined automatically by algorithm (please refer to official scikit-learn documentation, linked within Section 2.5).

e. Incorporates single hidden layer.

f. Within automated procedure, range 32–512 applicable to randomised search only (10–500 instead examined in Optuna).

g. Incorporates two hidden layers.

h. For each iteration of manual optimisation (only), parameter value adopted at layer 2 is identical to that corresponding in layer 1 (within automated protocols, the two are each fully independent).

prediction (i.e., local interpretability) [47]. Its essential function lies in assessing the influence that the removal of a feature holds upon the Shapley quantities assigned to those which remain. Unique SHAP values are in turn generated, describing the magnitude of the contribution held by features towards a prediction: strong positive or negative values indicating that the property has a definitive impact upon output–eliciting either definitive increase or decrease (respectively) in the measurement modelled. Such individual contributions may, if desired, be summed so as to create a perspective on global importance. Two primary variants of the SHAP algorithm have been reported: Kernel SHAP (model-agonistic), and Tree SHAP (applicable specifically to tree-derived techniques) [47,48]. Analysis of models was conducted through use of the SHAP Python package (v. 0.39.0)–adopting Tree SHAP within RF and XGB, and Kernel SHAP for all others.

## 2.7. Supplementation of QSAR uncertainty scheme in order to enhance relevance towards ML

The scheme for the evaluation of QSARs developed by Cronin et al., consisting of 49 assessment factors, was applied to ML-constructed models (as considered in general) [18]. Existing criteria were examined in order to ascertain which of those stood as insufficient with regards to the handling of concerns specific in deployment of ML. To facilitate this, factors were categorised in accordance with their relationship to the broad areas of model reproducibility, interpretability, and generalisability. Where appropriate, criteria were updated with supplementary guidance, so as to directly enhance ML-relevance.

## 3. Results and discussion

### 3.1. Impact of descriptor quantity upon model performance

It is known that selection of model features may have a key impact upon general predictive performance. The presence of excess, redundant, descriptors has tendency to introduce "noise" into a dataset [49–51]. For purposes of clarification, we define "noise" as representing elements within data which serve to confound and obstruct the detection of those legitimately meaningful trends and relationships–some of which may be of key importance in informing interpretability. The processes relating to feature reduction–by which the quantity of descriptors employed in training is lessened through the increasingly rigorous exclusion of collinear forms–are outlined within Section 2.2. Table 2 illustrates the dimensionality of the training data (in terms of descriptor numbers remaining) following application of each collinearity threshold–accompanied by the predictive quality of the models derived therefrom. By way of example, it is seen that unfiltered *T. pyriformis* (TH_Full) and rat lethality (LD_Full) datasets comprise 936 and 1087 descriptors respectively. With the exclusion of single features amongst those pairs exhibiting mutual collinearity of 90% or greater (TH_90, LD_90), these quantities fall to 447 and 546 (and so on).

As demonstrated within Table 2, the performance of all considered models (in this instance trained built using their default hyperparameter values) exhibited general increase in line with quantity of descriptors available for their construction. However, growth both in $R^2_{CV}$ (k = 10) and $R^2_{train}$ (properties each defined within Section 2.4), was seen to plateau far in advance of upper limit in feature number (~ 1000). Whilst slight variation was noted method-to-method, the latter metric could approach its maximal in data subsets containing fewer than 100 descriptors, and the former (as illustrated within Fig 2) in those composed of beneath 250. The observation relating to $R^2_{CV}$ is, of course, of greater practical relevance–indicating as it does that moderate exclusion of features can be undertaken with a minimal resulting impact upon

**Table 2. Predictivity of algorithms trained upon *T. pyriformis* growth inhibition/rat acute lethality parent and feature-reduced subsets (with corresponding descriptor totals relating to each further displayed).** Performance is presented in terms of $R^2_{train}$, $R^2_{cv}$ (k = 10) and $R^2_{over}$. Highest- and lowest-scoring models in former two metrics shaded green and orange respectively. Highest- and lowest-scoring in the latter (representing extent of overfitting) conversely coloured orange and green respectively.

| Data subset | n. Descriptors | Modelling approach and predictivity | | | | | | | | | | | | | | | | | |
| --- | --- | --- | --- | --- | --- | --- | --- | --- | --- | --- | --- | --- | --- | --- | --- | --- | --- | --- | --- |
| | | RF | | | SVM | | | k-NN | | | XGB | | | SNN | | | DNN | | |
| | | $R^2_{train}$ | $R^2_{cv}$ | $R^2_{over}$ | $R^2_{train}$ | $R^2_{cv}$ | $R^2_{over}$ | $R^2_{train}$ | $R^2_{cv}$ | $R^2_{over}$ | $R^2_{train}$ | $R^2_{cv}$ | $R^2_{over}$ | $R^2_{train}$ | $R^2_{cv}$ | $R^2_{over}$ | $R^2_{train}$ | $R^2_{cv}$ | $R^2_{over}$ |
| *T. pyriformis* (n. substances = 1,995) | | | | | | | | | | | | | | | | | | | |
| TH_Full | 936 | 0.965 | 0.751 | 0.214 | 0.899 | 0.758 | 0.141 | 0.796 | 0.681 | 0.115 | 1.00 | 0.757 | 0.243 | 0.929 | 0.767 | 0.162 | 0.956 | 0.800 | 0.156 |
| TH_90 | 447 | 0.964 | 0.750 | 0.214 | 0.902 | 0.746 | 0.156 | 0.782 | 0.660 | 0.122 | 1.00 | 0.778 | 0.222 | 0.953 | 0.792 | 0.161 | 0.968 | 0.806 | 0.162 |
| TH_80 | 256 | 0.964 | 0.748 | 0.216 | 0.903 | 0.742 | 0.161 | 0.776 | 0.652 | 0.124 | 1.00 | 0.776 | 0.224 | 0.961 | 0.779 | 0.182 | 0.969 | 0.802 | 0.167 |
| TH_70 | 150 | 0.964 | 0.740 | 0.224 | 0.895 | 0.726 | 0.169 | 0.758 | 0.618 | 0.14 | 0.999 | 0.758 | 0.241 | 0.959 | 0.748 | 0.211 | 0.969 | 0.781 | 0.188 |
| TH_60 | 101 | 0.961 | 0.726 | 0.235 | 0.885 | 0.716 | 0.169 | 0.754 | 0.613 | 0.141 | 0.999 | 0.748 | 0.251 | 0.955 | 0.731 | 0.224 | 0.971 | 0.768 | 0.203 |
| TH_50 | 69 | 0.961 | 0.722 | 0.239 | 0.873 | 0.720 | 0.153 | 0.755 | 0.625 | 0.13 | 0.998 | 0.748 | 0.25 | 0.943 | 0.745 | 0.198 | 0.971 | 0.767 | 0.204 |
| TH_40 | 35 | 0.961 | 0.719 | 0.242 | 0.822 | 0.700 | 0.122 | 0.756 | 0.609 | 0.147 | 0.992 | 0.725 | 0.267 | 0.871 | 0.709 | 0.162 | 0.943 | 0.732 | 0.211 |
| TH_30 | 18 | 0.944 | 0.600 | 0.344 | 0.633 | 0.552 | 0.081 | 0.682 | 0.513 | 0.169 | 0.966 | 0.585 | 0.381 | 0.639 | 0.528 | 0.111 | 0.748 | 0.569 | 0.179 |
| Rat acute oral lethality (n. substances = 8,448) | | | | | | | | | | | | | | | | | | | |
| LD_Full | 1,087 | 0.940 | 0.567 | 0.373 | 0.736 | 0.559 | 0.177 | 0.684 | 0.511 | 0.173 | 0.968 | 0.549 | 0.419 | 0.903 | 0.517 | 0.386 | 0.968 | 0.583 | 0.385 |
| LD_90 | 546 | 0.939 | 0.563 | 0.376 | 0.759 | 0.562 | 0.197 | 0.684 | 0.508 | 0.176 | 0.967 | 0.546 | 0.421 | 0.926 | 0.507 | 0.419 | 0.978 | 0.583 | 0.395 |
| LD_80 | 353 | 0.940 | 0.567 | 0.373 | 0.764 | 0.565 | 0.199 | 0.689 | 0.517 | 0.172 | 0.962 | 0.538 | 0.424 | 0.935 | 0.502 | 0.433 | 0.980 | 0.578 | 0.402 |
| LD_70 | 231 | 0.939 | 0.563 | 0.376 | 0.754 | 0.556 | 0.198 | 0.683 | 0.508 | 0.175 | 0.953 | 0.537 | 0.416 | 0.930 | 0.492 | 0.438 | 0.981 | 0.577 | 0.404 |
| LD_60 | 141 | 0.938 | 0.564 | 0.374 | 0.744 | 0.547 | 0.197 | 0.678 | 0.506 | 0.172 | 0.938 | 0.530 | 0.408 | 0.926 | 0.475 | 0.451 | 0.982 | 0.565 | 0.417 |
| LD_50 | 98 | 0.936 | 0.542 | 0.394 | 0.713 | 0.519 | 0.194 | 0.668 | 0.482 | 0.186 | 0.918 | 0.520 | 0.398 | 0.908 | 0.444 | 0.464 | 0.980 | 0.544 | 0.436 |
| LD_40 | 59 | 0.930 | 0.504 | 0.426 | 0.619 | 0.450 | 0.169 | 0.635 | 0.435 | 0.2 | 0.882 | 0.467 | 0.415 | 0.839 | 0.370 | 0.469 | 0.975 | 0.470 | 0.505 |
| LD_30 | 30 | 0.914 | 0.381 | 0.533 | 0.407 | 0.316 | 0.091 | 0.549 | 0.304 | 0.245 | 0.724 | 0.349 | 0.375 | 0.507 | 0.281 | 0.226 | 0.755 | 0.290 | 0.465 |

algorithm predictivity against test data. It is notable that near-identical performance in this metric is obtained through use of the TH_Full/TH_90 and LD_Full/LD_90 dataset variants–despite the reduced forms containing roughly half of the descriptor total of the unfiltered (an absolute deficit of approximately 500 in each instance). Indeed, even the shrinking of the descriptor pool by a factor of ten (as in TH_60 and LH_50) leads to a fall in $R^2_{CV}$ averaging only 0.04. A point shall inevitably come, nevertheless, whereby feature quantity falls to levels insufficient for capturing complexity inherent within the data. This may be exemplified through the markedly inferior performance of the TH_30 and LD_30 subsets (comprising 18 and 30 descriptors respectively).

Therefore, it is clear that an ideal balance must be struck between the twin concerns of model predictivity and transparency: if solely the former is sought, then feature exclusion may be deemed unnecessary. Within the sphere of toxicological QSAR (and elsewhere), however, it is likely that some degree of interpretability shall be considered desirable [52]. In this case, the sacrificing of descriptors, even at the cost of a minor drop in performance, may be judged an appropriate compromise (for a practical example of this, please refer to data presented within Section 3.5.2) [53]. With regard to the selection of features for retention/exclusion, calculation of pairwise correlations between descriptors is a standard approach. However, the decision of which feature to drop from a collinear pair may cause difficulty. A logical approach is to retain that which correlates most reliably to overall output. Nevertheless, it remains possible that descriptors not as statistically relevant towards the outcome, when considered individually, may yet have greater impact when modelling alongside an entire dataset [54].

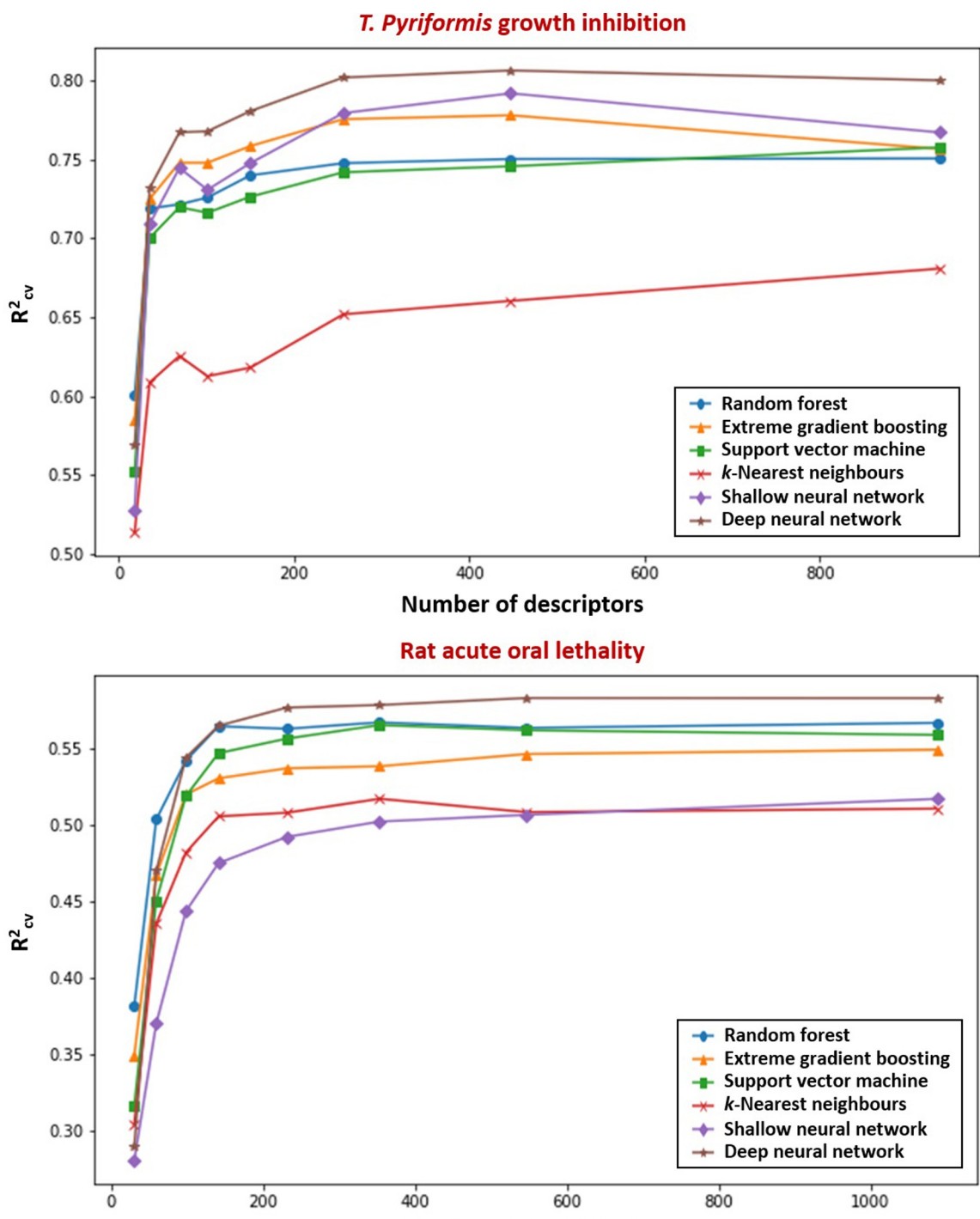

**Fig 2. Variation in predictivity of algorithms constructed upon *T. pyriformis* growth inhibition/rat acute lethality data subsets (expressed as R2CV, where k = 10), in accordance with quantity of training features present.**

## 3.2. Quality and consistency of training data

Although the general trends with respect to influence of descriptor quantity upon model performance (as described within Section 3.1) remained consistent across both datasets, prominent differences in absolute performance nevertheless separated the pair. As such, average disparity between corresponding *T. pyriformis* and rat lethality $R^2_{CV}$ (for example SVM in TH_90 and LD_90, or XGB in TH_70 and LD_70) stood at +0.212. Such notable variation can be accredited to the contrast present between the quality and consistency of both data collections, alongside the inherent complexity of the endpoint examined. Quality of input data (i.e., the associated error towards each datapoint) is a factor key in determining the capacity of algorithms to identify generalisable patterns–and thus by extension to formulate appropriate predictions relating to unfamiliar substances. Data curation and standardisation in form accordingly represent essential steps ahead of model training [55].

Considering the two data collections examined within this study, it is evident that the set representing *T. pyriformis* toxicity is the more amenable towards modelling application. *T. pyriformis* itself is a comparatively simple, unicellular organism–and mechanisms underlying the adverse effects of substances towards it are well characterised. Hydrophobicity-dependent narcosis (as modelled using logarithm of the octanol/water partition coefficient), and intrinsic chemical reactivity, are acknowledged in particular as being key influences [56–58]. The quality and consistency of the original data within this collection has been reported as high, acquired as it was following standardised procedures performed in a single laboratory, with experimental variability lying between 0.2–0.5 log units [59]. A rigorous curation workflow was followed in the course of its collation [21]. Furthermore, it contains few specifically acting compounds such as pesticides and pharmaceuticals–thus ensuring that a single mode of action (the aforementioned narcosis) is expected to predominate. By comparison, rat acute oral lethality stands as complex and poorly-characterised. This particular dataset had been compiled from a wide array of sources, and contains, in all, greater than 8,000 substances [23]. Due to its scale and the uncertain origins underlying many of the results within (issues discussed in detail by Karmaus et al., who quantified a margin of uncertainty of ±0.24 log units), the collection can be considered as carrying a higher degree of noise [60]. A broad range of chemical classes are covered, including natural products and specifically-acting substances such as pesticides–with generally limited knowledge present as relates to their potential modes and mechanisms of toxic action.

## 3.3. Cross-validation and the identification of model overfitting potential

An issue almost integral in supervised ML is that of overfitting–the phenomenon whereby a developed algorithm is attuned so acutely to the intricacies of its training set, that its subsequent capacity to generalise unseen data is compromised [61,62]. The presence of large deviation between training and test set predictive performance is indicative of an overfit model: one which displays weak generalisation ability, and one which accordingly generates external predictions of potentially questionable validity [63]. In order to simulate predictivity against non-training data in a rigorous manner, the technique of cross-validation is routinely employed [42]. As such, and in line with protocols described within Section 2.4.1, analysis was undertaken so as to assess the influence that quantity of folds (i.e., number of smaller sets into which the original had been divided) had upon performance of models as regards characterisation of unseen data. Fold numbers, k, ranging from two to 25 were investigated through each ML approach–with outcomes (expressed as $R^2_{CV}$ and inter-fold $R^2$ variability) illustrated in Fig 3. Across all methods, predictivity was poorer at lowest k values (i.e., k < 5). As k increased past

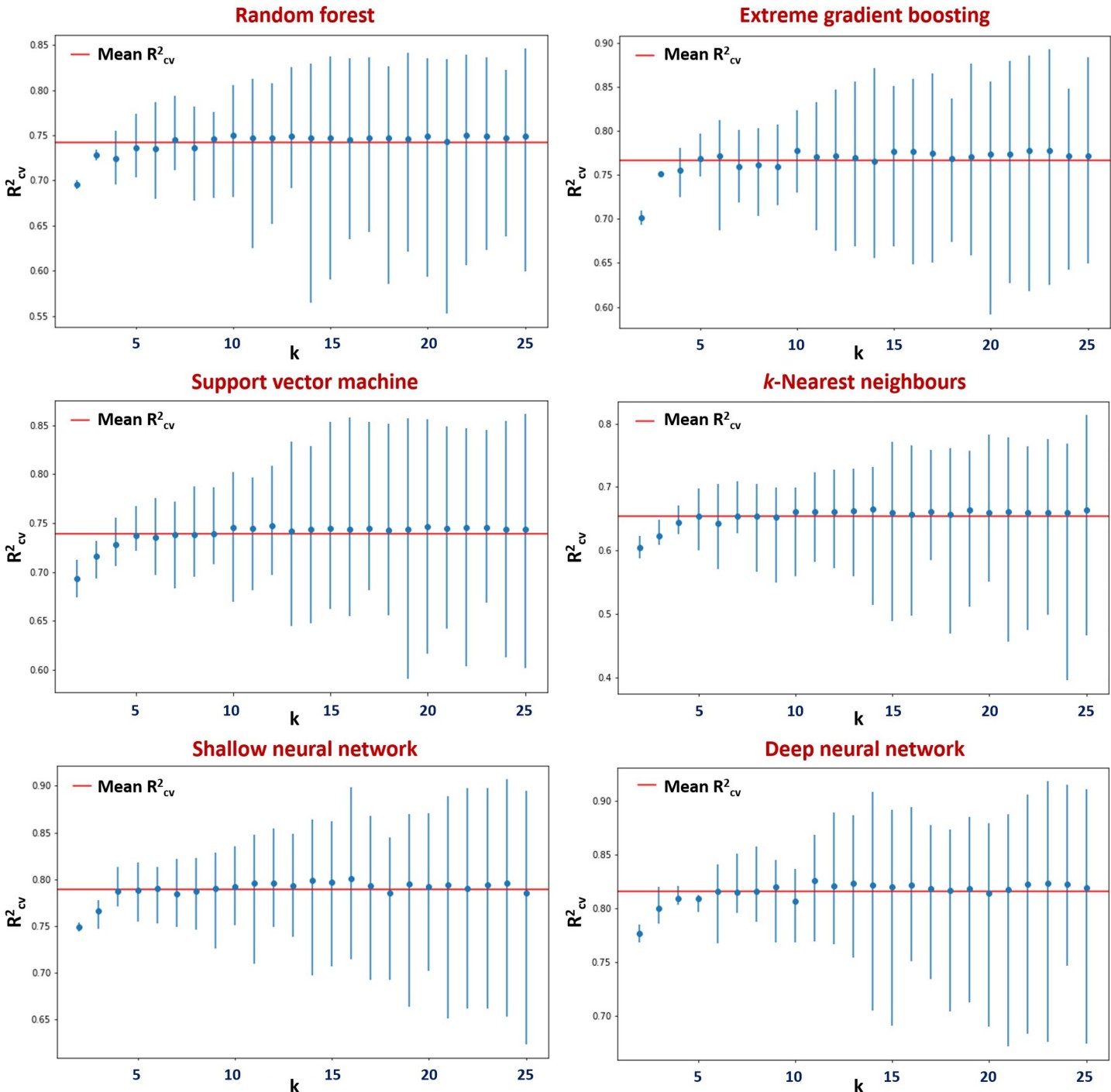

**Fig 3. Variation in parameters relating to apparent predictivity of models trained upon TH_90 subset, in accordance with number of folds, k, into which data was split ahead of cross-validation.** $R^2_{CV}$ describes average of $R^2$ values acquired from individual folds during their respective hold-outs. Ranges corresponding to maximal/minimal hold-out $R^2$ are, in each instance, displayed as vertical blue bars. Denoted in red is absolute mean of $R^2_{CV}$ values obtained, model-by-model, over all quantities of k.

five, performance was seen to rapidly approach its maximal. Further beyond, only variability in inter-fold $R^2$ (as represented by blue bars) about the mean was observed to grow.

When splitting, consideration must be given to the seeking of balance between two elements: firstly, that diversity within the dataset as a whole is appropriately captured within the training selection–and secondly that the hold-out (evaluation) set is large enough such that significant variation in the nature of the substances represented across each division does not arise [64]. Mean performance is influenced generally by the former factor, and degree of variance by the latter. Results from this exercise indicate that ten-fold validation tends to provide an optimal medium between the pair–combining maximum predictivity with an acceptable degree of variability. This observation accords with common practice noted within literature, whereby k = 10 is often employed (and furthermore noted for production of statistically unbiased outcomes) [65]. Although not explored through this study, the potential merits of worst-case training-validation split approach for application in toxicological QSAR (granting a conservative estimate of model predictivity) are acknowledged [66].

The extent to which each modelling technique, and data subset, exhibited liability towards overfitting was inferred through determination of $R^2_{over}$ (as defined in Section 2.4.2). These values are presented in the appropriate columns within Table 2. Whilst the magnitude of the figure exhibited no consistent relationship with subset size within-dataset (instead varying more reliably between respective ML approaches), a mean disparity between analogous *T. pyriformis* and rat lethality models was again discernible–standing at -0.146 (thus indicating greater tendency towards overfitting in the latter). Again, as considered within Section 3.2, this is attributable to the deviation in general data quality and uniformity between each collection: the enhanced level of noise present within the rat lethality set impeding the detection of appropriate patterns necessary for development of a broadly-applicable algorithm. As an illustration, the contrast between the magnitude of the average drops in $R^2_{CV}$ and $R^2_{train}$ from *T. pyriformis* to the parallel rat lethality models (a moderate 0.212 and a small 0.0664, respectively) should be noted.

## 3.4. Hyperparameter optimisation for enhancement of model predictivity

Optimisation of model hyperparameters was undertaken using three alternative techniques, with details relevant to each (including a summary of underlying theory) described within Section 2.5. Default values as present within the software packages applied were adopted in order to provide a performance baseline. Parameter influences upon predictivity were subsequently explored through simple manual adjustment–then by means of two automated approaches: the first being the scikit-learn randomised search function (*RandomizedSearchCV*), the second the Bayesian-rooted Optuna algorithm [43].

Impact of these procedures upon the performance of models trained through use of the TH_90 data subset is presented within Table 3. Metrics describing absolute predictivity of "default" algorithms are first listed, with the strength of the "optimised" forms expressed relative to these. Corresponding hyperparameter quantities themselves may be found listed within Table 1 –with additional details relating to each of the tuning processes accessible within S2 File (manual), S3 File (randomised search) and S4 File (Optuna). Results indicated that the more computationally-intensive protocol tended to provoke the stronger improvement in external predictivity: Optuna outperforming randomised search, and both besting manual efforts (average increases in $R^2_{CV}$ across models, relative to default, standing respectively at 0.0283, 0.0252 and 0.0183). The simplicity of the manual optimisation routine adopted, by which the influence of variation in each parameter from default was considered only in isolation, ensured that favourable alignments resulting from the concurrent alteration of two or more were to remain unexplored. Randomised search was able to offer advances on this–permitting all parameters to be tuned simultaneously, and hence widening the areas of accessible

**Table 3. Influence of hyperparameter optimisation protocols upon predictivity (k = 10) of models trained using TH_90 data subset.** Metrics (absolute) relating to the performance of algorithms developed through default hyperparameter sets are described. Relative variations within these, emerging post-optimisation, are further presented: values shaded green indicate favourable impact, and orange unfavourable.

| Optimisation approach | Performance metric | Modelling approach | | | | | |
|---|---|---|---|---|---|---|---|
| | | RF | SVM | k-NN | XGB | SNN | DNN |
| | | *Absolute predictivity (default hyperparameter sets)* | | | | | |
| None (default) | $R^2_{train}$ | 0.964 | 0.902 | 0.782 | 1.000 | 0.953 | 0.968 |
| | $R^2_{CV}$ | 0.750 | 0.746 | 0.660 | 0.778 | 0.792 | 0.806 |
| | $R^2_{over}$ | 0.214 | 0.156 | 0.122 | 0.222 | 0.161 | 0.162 |
| | $MSE_{CV}$ | 0.271 | 0.276 | 0.368 | 0.241 | 0.225 | 0.209 |
| | $RMSE_{CV}$ | 0.521 | 0.526 | 0.606 | 0.490 | 0.475 | 0.458 |
| | $MAE_{CV}$ | 0.378 | 0.363 | 0.441 | 0.354 | 0.339 | 0.317 |
| | | *Optimisation-induced variation in predictivity (relative to default)* | | | | | |
| Manual | $R^2_{train}$ | -0.002 | 0.072 | 0.005 | -0.026 | 0.01 | -0.001 |
| | $R^2_{CV}$ | 0.003 | 0.048 | 0.034 | 0.022 | -0.007 | 0.01 |
| | $R^2_{over}$ | -0.005 | 0.024 | -0.029 | -0.048 | 0.017 | -0.011 |
| | $MSE_{CV}$ | -0.003 | -0.053 | -0.036 | -0.025 | 0.009 | -0.01 |
| | $RMSE_{CV}$ | -0.003 | -0.054 | -0.03 | -0.025 | 0.008 | -0.012 |
| | $MAE_{CV}$ | -0.002 | -0.037 | -0.025 | -0.019 | -0.01 | -0.005 |
| Randomised search | $R^2_{train}$ | 0.002 | 0.071 | 0.063 | -0.014 | 0.028 | 0.019 |
| | $R^2_{CV}$ | 0.003 | 0.058 | 0.036 | 0.03 | 0.008 | 0.016 |
| | $R^2_{over}$ | -0.001 | 0.013 | 0.027 | -0.044 | 0.02 | 0.003 |
| | $MSE_{CV}$ | -0.003 | -0.063 | -0.04 | -0.033 | -0.008 | -0.016 |
| | $RMSE_{CV}$ | -0.004 | -0.065 | -0.033 | -0.034 | -0.009 | -0.018 |
| | $MAE_{CV}$ | -0.003 | -0.044 | -0.031 | -0.028 | -0.019 | -0.011 |
| Optuna | $R^2_{train}$ | 0.002 | 0.075 | 0.063 | -0.005 | 0.015 | 0.024 |
| | $R^2_{CV}$ | 0.003 | 0.058 | 0.036 | 0.033 | 0.017 | 0.023 |
| | $R^2_{over}$ | -0.001 | 0.017 | 0.027 | -0.038 | -0.002 | 0.001 |
| | $MSE_{CV}$ | -0.003 | -0.063 | -0.04 | -0.036 | -0.017 | -0.023 |
| | $RMSE_{CV}$ | -0.004 | -0.065 | -0.033 | -0.037 | -0.019 | -0.027 |
| | $MAE_{CV}$ | -0.003 | -0.044 | -0.031 | -0.03 | -0.025 | -0.016 |

space. A key shortcoming of this methodology lies precisely in its randomness: whilst favourable combinations are likely to uncovered, it is further possible that many may be overlooked. Through integration of Bayesian principles, Optuna offers a means of addressing such concerns–enabling focused examination around already-promising parameter arrays, as identified across prior iterations [45]. Although complexity of each approach increases in comparison to that of the previous, both time and degree of expert judgment required for application reduces.

Not all algorithms saw equivalent performance growth: although SVM $R^2_{CV}$ increased substantially by 0.058 following application of each automated approach, corresponding operations upon RF saw the metric barely influenced (advanced only by 0.003). Whilst any upturn within $R^2_{CV}$ (however small) may at first sight be welcomed, it is necessary to draw attention to those increases in overfitting potential (signified by heightened $R^2_{over}$) seen to accompany optimisation in SVM, k-NN, SNN (this in randomised search only) and DNN. Clearly, these emerge as growth in $R^2_{trsin}$ outstrips that of $R^2CV$–in turn signifying that capacity to fit the training data is simultaneously subject to improvement (a liability applicable in principle to all ML techniques). Therefore, as promising as these automated procedures undoubtedly are in

perfecting algorithm predictive aptitude ($R^2_{CV}$), this as a side-feature should nevertheless be borne in mind during their use.

## 3.5. Feature importance and model interpretability

The capacity to soundly interpret results obtained from a QSAR is essential to ensuring confidence in its trustworthiness and validity. As such, "Mechanistic Interpretation" is present as one of five original OECD Principles for the Validation of QSARs for Regulatory Assessment [13]. Put briefly, mechanistic interpretability rests upon capacity to define a pathway of causality linking molecular properties (i.e. features) and modelled endpoint [52,67]. It is recognised, however, that the nature of those methods adopted in ML lies generally at odds with the desire that their outputs be readily understandable and rationalisable by humans [68]. Indeed, many such techniques produce algorithms which are in effect a "black box"–their inner workings hidden to the user, and the reasoning underlying their predictions largely opaque [69].

Elucidation of those features most important in influencing output is highly desirable–yet in many instances challenging. The architecture of algorithms constructed through approaches such as RF and XGB is such that "model-specific" parameters, quantifying the relative contributions of the individual features, may by default be defined [70]. However, this is not so readily the case in those trained through use of alternative techniques (for example, NN). Accordingly, corresponding universally-applicable "model-agnostic" methodologies have been sought [71,72]. Whereas the aforementioned RF/XGB "specific" quantities may offer only "global" interpretability (describing impact of a feature upon the performance of the model as a unit), it is possible that selected "agnostic" approaches may in addition support characterisation of interpretation at a "local" (i.e., individual prediction) level [73]. Ideally, both local and global perspectives should be examined–and as such, models were subject to distinct forms of feature importance analysis representative of each.

**3.5.1. Permutation feature importance.** An overview of the principles underlying PFI (an agnostic approach offering model interpretability at the global level) is presented within Section 2.6.1. The technique was applied to models trained, adopting Optuna-optimised hyperparameters, upon the TH_90 data subset–with the ten highest scoring features identified in each instance presented within Fig 4. Both tree-based ensemble methods (RF and XGB) reported the descriptor SpMax2_Bhm (representing the largest absolute eigenvalue of Burden modified matrix–n 2, weighted by relative mass), to be most influential. Indeed, seven of the nine further features displayed appeared common to both.

Examination of output relating to each of the four additional models revealed routine presence of electrotopological state indices (exemplified by SHCsatu, SHdsCH and SHCHnX) [74]. As such, it can be inferred that alignment with respect to both topological and electronic properties constitutes a factor key in influencing the toxicological potency of molecules within this endpoint. However, notable is the fact that the typical returned values of "mean accuracy decrease", relating to those most important features, were discernibly higher within both RF and XGB than within SVM, k-NN, SNN and DNN. This indicated that the apparent impact of even the most "important" descriptors towards performance in these latter four models was, effectively, minimal. In any case (even as relates to RF and XGB), the means through which correlations between the identified features and observed toxicity might be rationalised, in a mechanistic sense, remain unclear. It should further be noted that a report by Hooker and Mentch advocated against the use of traditional permutation importance methods, arguing that they may be liable to produce misleading results–particularly when handling collections of inter-correlated features [75].

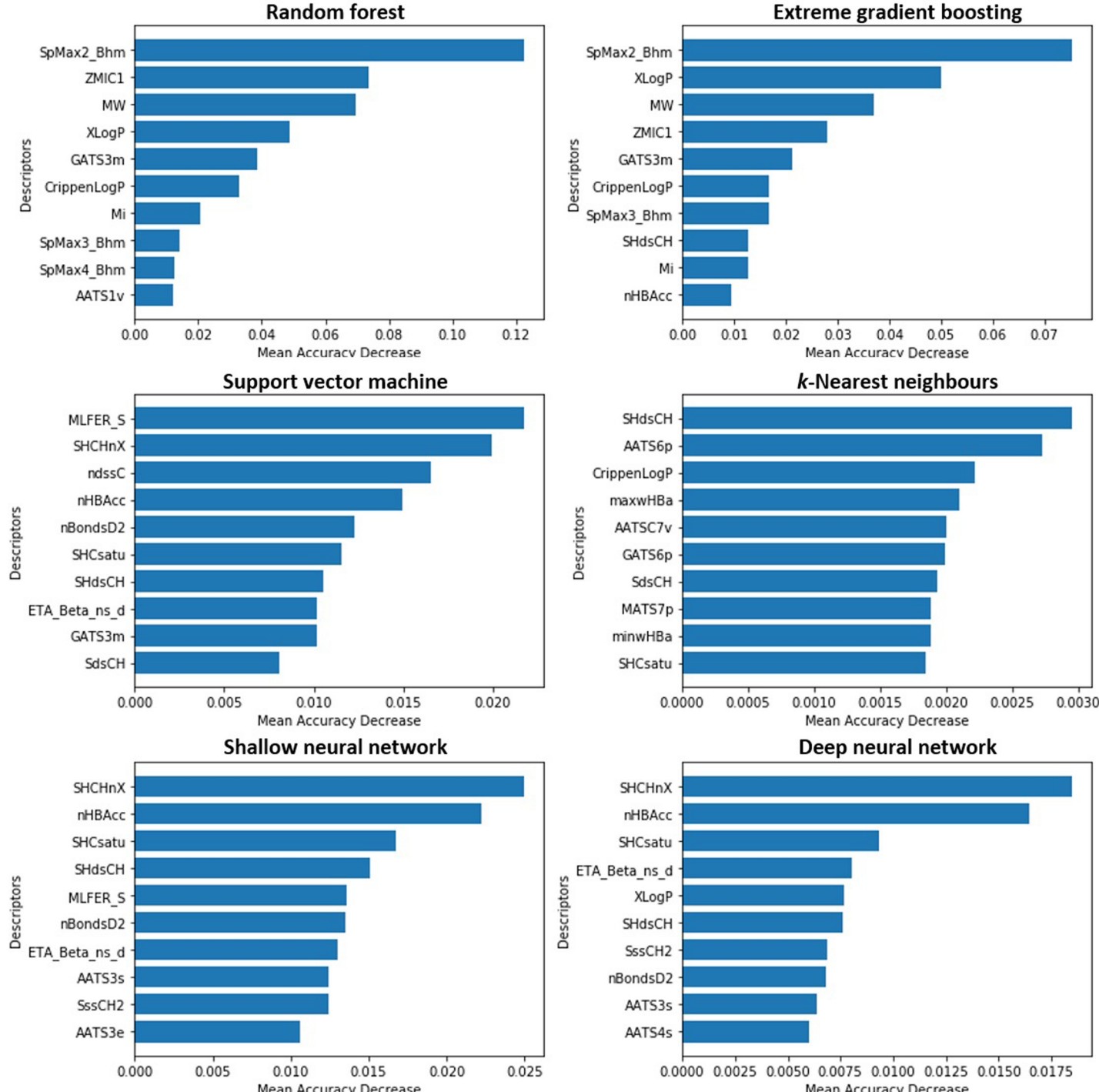

**Fig 4. Ordered listing of those ten features most prominent in influencing predictions issued by algorithms trained upon TH_90 data subset, as identified through application of permutation feature importance.**

**3.5.2. SHapley Additive exPlanations (SHAP).** The theory upon which SHAP methodology rests is outlined within Section 2.6.2. In practical terms, a key area of variance between it and classical PFI lies in the capacity to produce output which is locally interpretable. Summing

of feature contributions across all predictions can, nevertheless, be used to present a global per-spective as regards their significance [76]. Fig 5 provides illustration of the ten most-relevant descriptors, model-by-model (TH_90 data subset), as identified through SHAP. Within these plots, points correspond to individual predictions: position on x-axis indicates extent to which the feature has served either to increase (positive direction) or else decrease (negative direction) the absolute numeric value of the modelled characteristic. Colour scale describes the influence that the magnitude of the descriptor value itself holds upon the output quantity. By way of example, it can be seen, within the RF algorithm trained, that it was generally the case that a lower (blue) molecular weight (MW) was associated with reduced predicted toxic potency. For simplified expression of the generalised global importances of these features within their respective models, please refer to S5 File. Whilst it is clear that the level of insight offered by SHAP is considerable, it is nevertheless the case that methodological imperfections have been highlighted: some associated broadly with the application of Shapley values in the field of ML, and some concerning more the specifics of their implementation within the package [77,78]. The extent of overlap between features identified concurrently through SHAP and through PFI is variable across models–with figures extending from those in XGB (ten shared from ten) and RF (nine from ten), through to SVM, SNN and DNN (six) and finally *k*-NN (none).

Within RF and XGB it was further noted that identified descriptors possessed a definitive influence upon the majority of predictions offered–that is, that SHAP values other than zero were routinely observed (as is best exemplified through SpMax2_Bhm and MW). By contrast, it was common within those four alternatives (SVM, *k*-NN, SNN and DNN) that the great bulk of predictions instead registered SHAP scores of zero. This again suggested that contribution of these "most important" descriptors was, within such models, largely negligible–effectively mirroring the pattern as discerned through PFI. The apparent centrality of these features to performance in non-ensemble approaches appeared to arise only as a function of their impression upon a small proportion of predictions–an observation hypothesised as owing itself to the presence of an excessive quantity of inter-related descriptors, each serving to minimise the effective contribution of the other.

In line with the concept of predictivity-transparency balance, as introduced within Section 3.1, modelling through a reduced descriptor pool (shedding closely collinear features) was proposed as a route through which greater clarity might be achieved. Accordingly, the aforementioned analysis was repeated using the more compact TH_50 data subset: this being composed of 69 descriptors (as opposed to the 447 of TH_90). Fig 6 depicts the outcomes of this exercise, again displaying those apparent ten most-relevant features per each model (simplified expressions once more available within S5 File). It can be observed that such reduction had the intended effect of increasing the proportion of predictions definitively influenced by the identified descriptors (i.e., holding non-zero SHAP values)–thus indicating general greater engagement of the retained features. Further contrast may be noted in the consistency of descriptors returned–this being higher within the TH_50 cohort. As illustration, it was observed across all six models that three features were to occupy positions amongst those top four most-prominent: ETA_Alpha (sum of alpha values of all non-hydrogen vertices), nHBAcc (number of hydrogen-bond acceptors) and AMW (average molecular weight).

ETA_Alpha functions essentially as a metric of molecular polarisability–a property which itself correlates strongly alongside lipophilicity [79]. SHAP output suggests, in each instance, that increase in the value of this parameter notably drives concurrent increase in predicted toxicity (and vice versa). This would accord with accepted understanding of the key determinants of acute lethality in *T. pyriformis*–which posits extent of hydrophobicity to be perhaps most dominant of all [56–58]. The prominence of nHBAcc, and its inversely-proportional

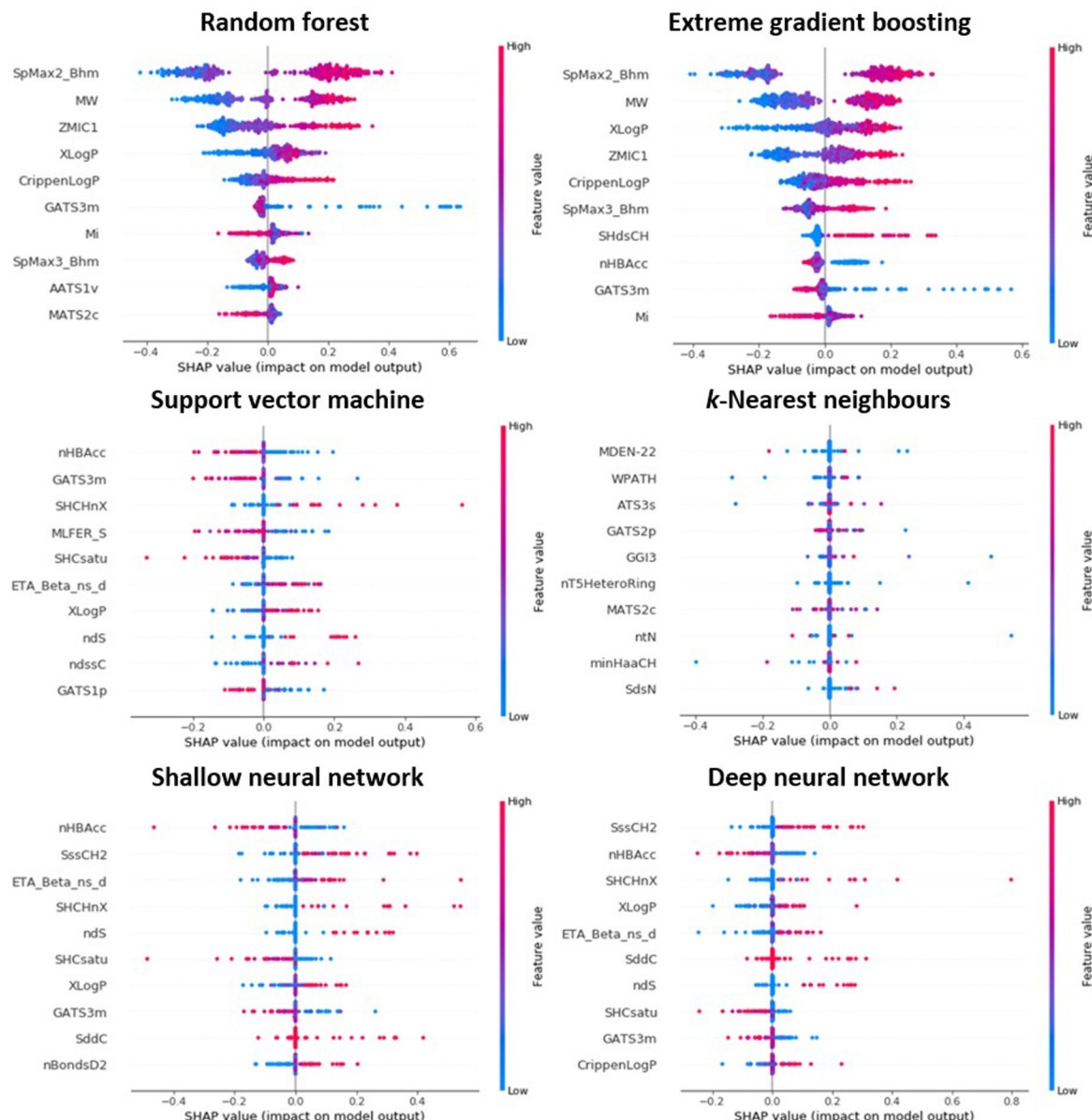

**Fig 5. Ordered listing of those ten features most prominent in influencing predictions issued by algorithms trained upon TH_90 data subset, as identified through application of SHAP (please refer to S5 File for corresponding simplified expressions of generalised global feature importance).**

relationship with estimated toxicity, can be interpreted as further representing this association: hydrogen-bonding itself holding great influence upon tendency towards solubility in polar medium. Promising as the identification of this link may be, it is important to draw attention to the fact that these features account only (in a definitive sense) for toxic outcomes arising through means of narcosis. As such, mechanisms mediated by way chemical reactivity, or else specific receptor interactions, remain unaccounted for [58,80]. Furthermore, acquisition of this insight came at the expense of model performance (average $R^2_{CV}$, TH_90 = 0.750; TH_90 = 0.721)–effectively illustrating management in the balance between transparency and predictivity.

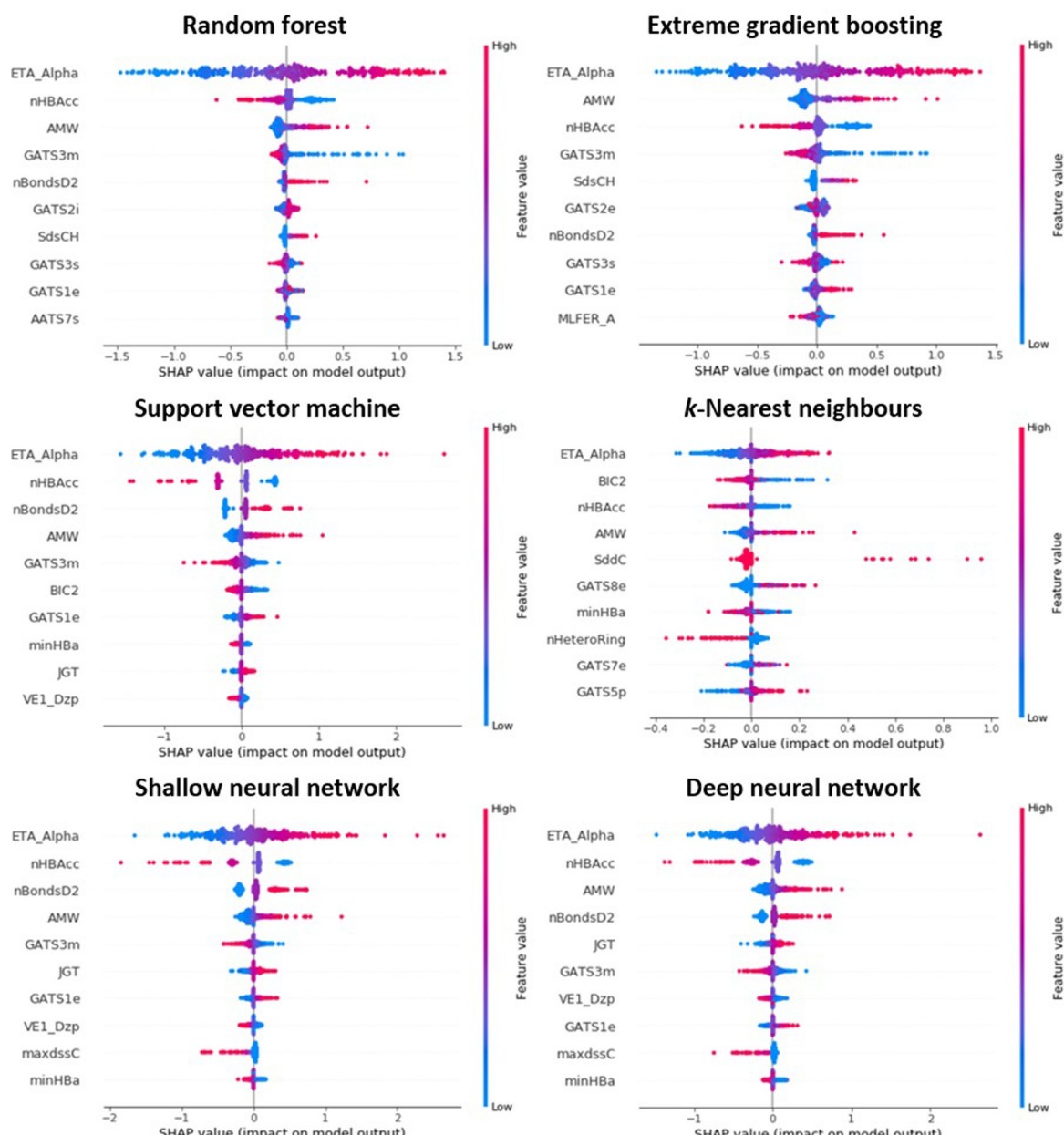

**Fig 6. Ordered listing of those ten features most prominent in influencing predictions issued by algorithms trained upon TH_50 data subset, as identified through application of SHAP (please refer to S5 File for corresponding simplified expressions of generalised global feature importance).**

## 3.6. Assessment of uncertainty relevant within machine learning-derived QSAR models

The identification and characterisation of uncertainties associated with predictions acquired from a toxicological QSAR assists in demonstration of its acceptability for a given purpose of interest [12,81]. A series of 49 assessment criteria, summarising the primary characteristics of such models as relates to their creation, description and application, were developed by Cronin et al., with intention of facilitating uncertainty evaluation [18]. It is acknowledged, however,

that in many regards, ML presents challenges lying beyond the remit of the guidelines as currently expressed. As such, we set out to evaluate the criteria in light of their suitability towards assessment of ML models–identifying areas within which their amendment and extension might be beneficial in enhancing general applicability. Three elements in requirement of particular attention were recognised, these being: reproducibility, interpretability and generalisability. Criteria relevant to each of these aspects were updated–with both the nature of the alterations and the rationale underlying them discussed forthwith. It must be stressed that such suggestions do not aim to discredit the ability of the scheme in its current state to evaluate ML models, instead they provide recommendations for further analysis that ensures all aspects of model uncertainty are understood by both the developer and user.

**3.6.1. Reproducibility.** Key to the validity and reliability of any experimental process is the assurance that its procedures, and by extension its outcomes, are each readily replicable [82,83]. To ensure the reproducibility of models, sufficient documentation is required, through

**Table 4. List of those assessment criteria for individual areas of uncertainty, variability or bias within toxicity-prediction QSAR (as presented by Cronin et al. [18]) updated in light of consideration of concerns specific to application of ML.** Each is grouped in accordance with its relevance either to the reproducibility, interpretability or generalisability of models. Updates to text under heading "comment or other information" are displayed in italics. Please refer to S6 File for presentation in context of unabridged scheme.

| ID | Assessment criteria | Comment or other information |
|---|---|---|
| **Reproducibility** | | |
| 2.1a | Definition and description of model (related to assessment criterion 3.1a) | All terms e.g., descriptors, statistical values, *hyperparameters and ranges*, algorithms should be defined. The QMRF is a possible reporting format. |
| 2.1c | Transparency of the model | *A transparent model can be reproduced, and the model output is (reasonably) interpretable, i.e., user can understand the causation of a prediction.* |
| 3.1a | Reproducibility of the model or QSAR (related to assessment criterion 2.1a) | To determine reproducibility, the model is assumed to be transparent (see assessment criterion 2.1c). *Source code should be provided, with computational infrastructure detailed.* |
| 3.1b | Reproducibility of the QSAR prediction | To obtain reproducible predictions, all parameters (descriptors) need to be available and controllable. *Seeds to control randomisation for certain algorithms need to be specified.* |
| **Interpretability** | | |
| 2.1c | As above | As above |
| 2.4c | Relevance of descriptors to mechanism of action/ AOP | *Feature importance techniques should be used for algorithms that employ large quantities of descriptors, relating highest scoring descriptors to the mechanism.* |
| **Generalisability** | | |
| 1.5a | How appropriate is the modelling approach for the endpoint and to deal with the complexity/non-linearity of the data | This requires a pragmatic and subjective assessment, e.g., a data set based on one mechanism with a single overriding descriptor can be modelled more simply than a more complex scenario. *If applicable, both the optimisation procedure and the sufficiency of resulting approach complexity should also be considered.* |
| 2.2a | Statement of statistical fit, performance and predictivity | The use of appropriate validation methods, *resampling techniques*, and/or external test sets should be demonstrated, different metrics may be required for different models. |
| 2.2b | Interpretation of statistical fit etc with respect to biological measurement error and variability | *The use of strategies to limit overfitting (e.g., early-stopping, pruning, regularisation) may be required for certain algorithms.* |

which unambiguous definitions of all amenable factors (as adopted in development) are presented. In spite of this, detailed reporting of methodology and output has often been overlooked within ML and artificial intelligence–with such issues only recently gaining attention [84]. It is apparent that ML presents a unique series of challenges which must be addressed when seeking to guarantee replicability. Commonly-employed techniques tend to incorporate large numbers of freely variable hyperparameters, each of which may typically be tuned independently of the other. Default values are themselves liable to be inconsistent–dependent not only upon the identity of the software employed, but also on its particular version or implementation form [85]. Furthermore, intrinsic to the training of many ML algorithms is presence of randomness: notably within NN, whereby connection weights are assigned stochastically [86]. Without control through consistent application of pseudo-random number generator seeds, reproduction of such models is a practical impossibility.

For non-ML QSARs, the provision of details concerning statistical technique, composition of training/ test sets (alongside molecular descriptors) and performance is generally sufficient in order to ensure confidence in reproducibility. However, given the aforementioned considerations exclusive to ML model construction, it is evident that an expanded range of definitions shall be required such that like demands may be satisfied. Accordingly, in addition to descriptor sets and those details relating predictive performance, it would be necessary to state both the hyperparameter quantities and the values of any seeds adopted in random number generation. Identification of computational software (incorporating version/implementation specifics) and hardware shall likewise be a requirement–whereas for maximal transparency, provision of full source code is desirable. Prior consideration has been granted to the development of schemata intended to promote practices facilitating ML replicability [82,87–89]. It is with further reference to such efforts that we were able to suggest updates to four of the 49 uncertainty assessment criteria presented by Cronin et al. [18]: changes which are themselves outlined within Table 4 (please refer to S6 File for presentation in their unabridged form).

**3.6.2. Interpretability.** Interpretability of QSAR is dependent upon identification of plausible relationships associating variability in molecular properties with ultimate predictive outcome [52,67]. For reasons introduced within Section 3.5, this task may prove particularly demanding where ML is considered. Traditional QSAR is characterised by dominance of simpler "model-based" techniques (such as linear regression), founded upon parametric statistical assumptions and, as such, considerate of processes underlying output variance [90]. Typically trained upon comparatively small datasets, these employ a minimal quantity of descriptors. By contrast, ML methodologies may be classified as "model-free"–working without underlying assumption, and concerned solely with acquisition of optimal predictivity. There exist few limitations with regards to potential complexity in algorithms constructed through these means. A multitude of features may be employed in their training, thus ensuring that preliminary identification of those holding utmost influence is a necessity. Within Sections 3.5.1 and 3.5.2, two agnostic methodologies intended for evaluation of the importance of features are explored: one offering global interpretability, the other local. It should be noted that alternative forms of feature importance-detection exist–including those offering direct insight in terms of molecular structural properties, and those otherwise specific to particular techniques [91]. However, a full consideration of their merits and shortcomings sits beyond the scope of this study.

Although ML models can indeed be reasonably interpreted through employment of the aforementioned approaches, practical success in doing so remains contingent upon two prerequisites. Firstly, those descriptors holding meaningful influence upon the functioning of the algorithm must be definitively identified [49,51,92]. If necessary, feature reduction should be performed (even at minor expense in terms of overall predictive power), such that the confounding influence of collinear relatives may be mitigated [53,54]. With the exception of those

trained through tree-based ensemble techniques (RF and XGB), our models exhibited a tendency to display high sensitivity towards the obscuring effect of large descriptor pools upon the apparent importance of individual features. In these instances, only when pool size had been sufficiently minimised was interpretation able to be attempted.

The second factor relates both to the availability and appropriate application of knowledge relating given descriptors, in a mechanistic sense, to an endpoint of interest. With respect to the adopted *T. pyriformis* endpoint and associated substance collection, such knowledge exists in an established form–and could accordingly be drawn upon in order to facilitate the framing of prominent features (particularly through the smaller TH_50 data subset) in light of their driving of hydrophobicity-associated baseline, narcotic toxicity [56–58]. Of course, were such clear understanding not to exist, then the outcomes of feature importance analysis would represent only correlation–with the existence of any causal linkage remaining a matter for speculation. It is further possible that more practically-interpretable descriptors may yet offer meaningful insight, even if they should fall outside of the list of features ranked as most prominent (in which case they may risk being overlooked). Considering all points, two relevant Cronin et al. criteria were deemed appropriate for update: these being presented within Table 4 and (in unedited form) S6 File [18].

**3.6.3. Generalisability.** The principles of generalisation and model overfitting are considered within Section 3.3. Causes underlying the emergence of this phenomenon have been classified by Ying as taking one of three primary forms [16]. The first may be considered a product of the learning of noise within the training set–a concept defined with Section 3.1, and in this instance relating to detection of specific trends in the data which, although not of universal relevance, may later inform external prediction. As might be anticipated, algorithms within this study trained upon the "noisier" rat lethality dataset were more inclined to overfit (that is, they displayed greater $R^2_{over}$), when related to those developed using the more homogenous and precisely-curated *T. pyriformis* inventory. Secondly, it is the case that particularly complex models (such as those composed from an excessive quantity of features) may display imbalance in favour of variance over bias, culminating in an increased accuracy concurrent with reduced broad consistency. The final factor relates to the routine employment of multiple comparison procedures during the process of training. It is perhaps inevitable that this shall, in time, result in the selection of parameters which have no positive impact upon general model performance.

On account of their considerable potency as relates to pattern recognition (and accompanying sharp focus as regards optimisation of predictive performance), the liability of ML techniques towards the introduction of overfitting is generally far greater than those of classical approaches (which may again be exemplified by linear regression). Accordingly, whilst examination of test set performance through cross-validation is considered best practice in both instances, additional attention has been placed upon the formulation of methodologies which aim to promote the training of more generalisable ML algorithms. Amongst these are concepts including early-stopping, regularisation and drop-out (the latter of the three being specific to NN and their derivatives) [16,93,94]. As such, the scheme of Cronin et al. may once more be revised (please refer to Table 4 and S6 File for details regarding those three criteria amended) [18].

## 4. Good practice in application of machine learning for development of predictive toxicology QSAR

The evaluation of ML as applied in toxicity prediction, combined with consideration of the uncertainties associated, has enabled the identification of aspects of good practice which

should be adhered to in order that the acceptability of such models (particularly with regards to the supporting of chemical safety assessment) might be improved.

These may be summarised:

- Biological data to be modelled should be evaluated in terms of quality, consistency, coverage of mechanisms etc.

- The outcomes of assessment of the biological data should be used to assist in problem formulation, particularly in provision of realistic (and not-overoptimistic) performance targets.

- Well-performed feature selection is necessary to reduce noise and collinearity. Fewer descriptors are also likely to assist in interpretability. If feature selection is not performed, then some rationale should be stated as to why.

- Descriptors must be appropriate for purposes of modelling of the effect, i.e., they must relate in some way to the putative mechanisms of action. It is accepted that for large datasets, full mechanistic definition is often likely to be implausible. In such instances, the approach and descriptors utilised should be justified and interpreted as best possible.

- Once constructed, the qualities of a model should be evaluated in terms of all aspects: predictive performance, interpretability and nature of any uncertainties.

- 10-fold splitting, or thereabouts, is optimal for use in cross-validation. Beneath this, model performance tends to be understated–a greater number, by contrast, adds little value.

- Model performance should be related to data quality i.e., to ensure the model does not fit beyond its limitations.

- Hyperparameters tuned during the optimisation procedure should be declared–with the approach undertaken in doing so appropriate for the quantity considered.

- Whilst selection of an ideal algorithm may be based upon performance metrics, the complexity and interpretability of a model should be considered, dependent upon its intended purpose.

- Interpretability of the model is crucial. Descriptors important in influencing output can be identified–with SHAP (in particular) offering a useful approach towards achieving this.

- Mechanistic interpretability does not automatically follow from identification of key descriptors. A direct relationship between the feature, and the means through which it contributes towards observed toxicity, should be established.

- Full documentation of the model should be provided, clearly demonstrating adherence to the good practice principles (as described above).

## Supporting information

**S1 File. Supplementary Material 1.** Influence of quantity of splits, k, upon properties including: ¤ Dataset fold size (i.e., average number of substances within). ¤ Apparent predictivity of models trained upon TH_90 data subset, expressed in terms of R2CV (average of R2 values acquired from individual folds during their hold-out), R2min and R2max (respective minimum and maximum in fold-wise R2). *Denotes parent dataset (prior to splitting) (XLSX)

**S2 File. Supplementary Material 2.** Hyperparameter optimisation (manual). Optimisation curves displayed within figures below outline influence upon the model performance metrics $R^2_{train}$ and $R^2_{CV}$ (k = 10), arising as a consequence of the incremental variation in value of a given hyperparameter over its defined range (with all others simultaneously held constant at their default quantities).
(DOCX)

**S3 File. Supplementary Material 3.** Hyperparameter optimisation (randomised search). Displayed within figures below are values of performance metrics $R^2_{train}$ and $R^2_{CV}$ (k = 10) relating to models generated through each trialled random hyperparameter combination.
(DOCX)

**S4 File. Supplementary Material 4.** Hyperparameter optimisation (Bayesian). Figures below illustrate aspects of the iterative Bayesian optimisation procedure enacted upon each model, courtesy of Optuna.
(DOCX)

**S5 File. Supplementary Material 5.** SHAP-determined **a**bsolute global feature importances relating to each model, as trained upon *T. pyriformis* TH_90 and TH_50 data subsets through adoption of Optuna-derived hyperparameter sets.
(DOCX)

**S6 File. Supplementary Material 6.** List of those assessment criteria for individual areas of uncertainty, variability or bias within toxicity-prediction QSAR (as presented by Cronin et al. [18]) updated in light of consideration of concerns specific to application of ML. Each is grouped in accordance with its relevance either to the reproducibility, interpretability or generalisability of models. Updates to text under heading "Comment or Other Information" are displayed in bold.
(XLSX)

## Author Contributions

**Conceptualization:** Samuel J. Belfield, Mark T.D. Cronin, James W. Firman.

**Data curation:** Samuel J. Belfield, James W. Firman.

**Investigation:** Samuel J. Belfield.

**Methodology:** Samuel J. Belfield, James W. Firman.

**Visualization:** Samuel J. Belfield, Mark T.D. Cronin.

**Writing – original draft:** Samuel J. Belfield, Mark T.D. Cronin.

**Writing – review & editing:** Mark T.D. Cronin, Steven J. Enoch, James W. Firman.

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
