## [Decision Letter · Decision Letter 0]

31 Oct 2022

PONE-D-22-28105Good practice for machine learning methods in predictive toxicologyPLOS ONE

Dear Dr. Cronin,

Thank you for submitting your manuscript to PLOS ONE. After careful consideration, we feel that it has merit but does not fully meet PLOS ONE’s publication criteria as it currently stands. Therefore, we invite you to submit a revised version of the manuscript that addresses the points raised during the review process.

We look forward to receiving your revised manuscript.

Kind regards,

Hilal Tayara

Academic Editor

PLOS ONE

Additional Editor Comments:

- I recommend the authors share the public and curated datasets in this study and the source code for reproducibility. 

Reviewers' comments:

Reviewer's Responses to Questions

**Comments to the Author**

1. Is the manuscript technically sound, and do the data support the conclusions?

Reviewer #1: Partly

Reviewer #2: Yes

Reviewer #3: Yes

2. Has the statistical analysis been performed appropriately and rigorously? 

Reviewer #1: Yes

Reviewer #2: Yes

Reviewer #3: Yes

3. Have the authors made all data underlying the findings in their manuscript fully available?

Reviewer #1: Yes

Reviewer #2: Yes

Reviewer #3: No

4. Is the manuscript presented in an intelligible fashion and written in standard English?

Reviewer #1: Yes

Reviewer #2: No

Reviewer #3: Yes

5. Review Comments to the Author

Reviewer #1: I like the general tone of this paper to provide the causal interpretation of machine learning models for toxicology and demonstrating that the current level is far away from needed. If the goal of this paper is to present the status of machine learning (ML) methods in toxicology relative to the traditional regression analysis, then this paper satisfies it. If the goal is deeper to give a guidance what needs to be done in machine learning to reach needed causal level in toxicology, then this goal is not reached from my viewpoint. Please clarify the goal. For the second goal it can be useful to get more specific considerations of inconsistency of the mechanistic/causal interpretation with ML models to get reasons of inconsistencies like dealing with attributes, which have low scores or too many attributes for humans to see causal relations. Detailed comments are below.

Fig 1.

Review. It is strange to see deep learning outside of supervised and unsupervised learning.

153. These data were reduced to 1,995 substances following the removal of duplicates.

363 Over 1,000 descriptors were calculated very rapidly in this study, as is common with most ML

368 This identified 936 significant descriptors for the Tetrahymena dataset, and 1,087

369 descriptors for the LD50 dataset

Review: Please provide in a single place the exact number of substances/cases and dimensions/descriptors of these data for each model and discuss the possible overfitting depending of the ratio between these numbers.

222 space between neighbours is measured by an appropriate distance metric.

232 NNs are taught through an iterative process, where

233 connection weights and biases between neurons are modified using a back-propagation

234 algorithm that aims to minimise the error function [39].

628 Certain ML methods, such as decision trees

629 and KNN, which researchers have used exhaustively, may already be classified as

630 interpretable, where the logical algorithm structure enable feature importance’s to be

631 deduced [48].

Review. Please make these and other summaries of ML methods deeper. Finding “appropriate distance metric” is an open problem for K-NN for decades. It is very sensitive to data scaling, and some distances have no meaning for heterogeneous data like blood pressure and temperature, which are common in ML. Please avoid following simplified sources and outline advantages and disadvantages of methods not only what they compute.

302 SHAP is a recently developed method at the cutting-edge of model interpretability originating

303 from the Shapley values of cooperative game theory [43]. Shapley values provide a unique

304 method to attribute a model’s outputs towards feature contribution, and therefore

305 guarantees the satisfaction of the three important properties: local accuracy, missingness,

306 and consistency [44]. SHAP values are assigned for each feature for individual predictions…

509 [61] advocated against traditional permutation importance methods, finding that they can

510 give rise to misleading results particularly while dealing with correlated features. Therefore,

511 to unearth stronger and more confident results for interpretability, an additional approach

512 defined as Shapley Additive exPlanations was undertaken.

527 from Shapley values that originate from coalitional game theory. This method enables the

528 pay-out (i.e., the prediction) to be fairly distributed among the players (descriptors); thus,

529 allowing the contribution each presents to be quantified [48].

Review. While some ML methods are presented without their pluses and minuses, SHAP is presented with this information, but unfortunately it is misleading. It must reflect critiques of SHAP in the ML literature, see [R1-R4]. In particular, SHAP brought from the cooperative game theory in economics the assumption that the evaluation/cost/award function in ML has the same nature as a monetary award in economics that must be split between players, which is expressed formally via additivity. This is at least questionable in ML for typical heterogeneous data [R1], which can make uniqueness of SHAP irrelevant for many ML tasks. You pointed out that permutation can be misleading and should do the same with Shapley Additive exPlanations and analyze its results critically too. You should justify “unearth stronger and more confident results” for it or make this statement softer.

435-36 Folds ranged from 2 to 25 were investigated with each ML algorithm. … For all ML methods, cross

437-38 validation demonstrated that the performance of all models was poorer with a low number

438 of folds i.e., up to five.

769 10-fold validation, or thereabouts, is optimal for cross validation. Less than this and

770 the model performance is adversely affected, more than this adds little value.

Review. Please provide the number of cases in each fold in experiments with different number of folds. While k-fold average is a typical way to evaluate the quality of the ML model, now the growing interest is associated with the worst-case splits to folds [R5], which is more appropriate for the task with the high cost of error, and it seems the some toxicology tasks are among those ML tasks.

489 understanding of which features are providing the greatest value to each algorithm. Hence,

490 gathering this information requires the calculation of the importance of features.

Review: while it is useful, it does not make the model interpretable for the end user/domain expert as you discuss in 570-576. It requires that the domain expert confirms the order of importance of features from the toxicology viewpoint, which can be spurious. Please comment more specifically is the order acceptable from the toxicology viewpoint. In essence a usability study is needed.

628 Certain ML methods, such as decision trees

629 and KNN, which researchers have used exhaustively, may already be classified as

630 interpretable, where the logical algorithm structure enable feature importance’s to be

631 deduced [48].

Review. It is widely accepted that decision trees and logical rules are interpretable. It is not clear for me why you do not present them in the paper analyzing only methods, which have difficulties to be interpreted. Their comparison in accuracy and interpretability with the decision trees will show if any of them have advantages. I feel that it should be done.

654-668 Review. Please make this part clearer. It is not clear what you want to say about the causal relations in the toxicology and the interpretability of the attributes. Do you want to say that of those attributes which are shown as a most informative in the algorithms give you a clue to explore a cause.

719-747 Review. As I already pointed out, you could be more cautious saying that linear regression models are fully interpretable. It is not always the case for the heterogeneous data [R1].

References

[R1] https://arxiv.org/pdf/2009.10221

[R2] David S. Watson, Conceptual challenges for interpretable machine learning, Synthese (2022) 200:65

https://link.springer.com/content/pdf/10.1007/s11229-022-03485-5.pdf

[R3] Watson DS. Rational Shapley Values. arXiv preprint arXiv:2106.10191. 2021 Jun 18.

[R4] Fryer D, Strümke I, Nguyen H. Shapley values for feature selection: The good, the bad, and the axioms. IEEE Access. 2021;9:144352-60.

[R5] C Recaido, B Kovalerchuk, Interpretable Machine Learning for Self-Service High-Risk Decision-Making, 2022 26th International Conference Information Visualisation, Vienna, https://arxiv.org/pdf/2205.04032

Reviewer #2: The study main aim is to provide a well formed framework to be considered as a best practice methodology for toxicology prediction. Different models were established with a variety of assessment methods to assure correct and efficient results. Model building also included the steps to construct feature importance which would be of great importance for easier and more efficient prediction of chemical compounds toxicity. The development of such models would guide the studies in the same domain with a clear workflow that was shown practically through this study.

I suggest considering the manuscript for publication but I greatly recommend considering the comments. Mainly, reforming many sentences is important to get the ideas easier. In addition, a more focused description of the ideas and direct link to the experimental results would be helpful to make the reader understand and get the benefit from the research.

- General English enhancements and corrections are recommended for more clear explanations of ideas. Generally, long sentences are frequent through the manuscript making it difficult to get the ideas.

- Please consider rewriting or clarifying the following sentences:

- 103 – 106 please rephrase a very long sentence.

- 158: very long and complicated, please make it more clear and easier to understand.

- 298: “In other words, the relationship…” this sentence needs to be clarified

- 408 “As can be seen in Fig 2, …”

- 424: “A certain degree of feature ..” please consider rewriting the sentence shorter

- 436: “ For all ML methods, cross-validation demonstrated …”

- 455:” Thus, with the results confirming conforming.. ”

- 483: “Descriptors utilised within ...”

- 614: “Assessment of uncertainty related toward..”.

- 697: “Therefore, to ensure that the assessment criteria are suitable for a variety of modelling approaches … “.

- 716: “Despite this, the use of such strategies may not guarantee…”

- The following comments are recommended to be considered:

- I would recommend adding more recent work about using ML in toxicology prediction from the literature.

- Additional details about dataset preprocessing would be useful

-

- 170: “Redundant descriptors were removed that were either uninformative or unstable.” Redundancy could be understood as repetitions, but on what basis you decided uninformative or unstable descriptors?

- 176: “Pairwise correlation coefficients values used to limit collinearity and create the subsets were: 0.9, 0.8, 0.7, 0.6, 0.5, 0.4, and 0.3. The descriptor of the pair that reported the weakest correlation to the target was omitted.”

o This is not completely clear, on what basis did you decide the correlation thresholds to drop descriptors correlated to the target?

o Also, you assumed complete independence between descriptors by doing this, which is not always correct.

- Please add the proper citation for the sci-kit learn library when mentioned.

- 218: Add reference.

- Many details in the results section is better to be moved to the data and methods section:

o 3.1: Analysis of Datasets

o 3.5: Many details in the subsections under the feature importance section.

o 3.6.1 Reproducibility: being a subsection within the results and discussion section, what are the results mentioned or discussed in this section, please make it clear, if any.

o In all subsections 3.6.1,3.6.2, 3.6.3, I would suggest reducing the theoretical description of each criteria to the least possible. I Also suggest that most of the derails in these sub-sections can be moved to the methods section while removing any redundancy. The subsections did not mention the results that prove how the suggested models satisfies these assessment criteria.

- 370: What was the basis on which you selected these collinearity levels?

- Please indicate clearly the subsets of data used for training and testing the ML models. Did you keep an independence test set? Or you only used cross-validation?

- 463: Could you highlight what were the hyperparameter values that resulted in the best performing models?

- In Table 4, please highlight the best results to be easier to find.

- Please show how optimization improved the prediction results by pointing out the difference between the model performance before and after optimization .

- What is your comment on the wide gap between training and testing R2 in most of the approaches, how was this gap affected by optimization.

- 580: “The uncertainty criteria were grouped into ten components that summarise the main characteristics of a QSAR …” please add few details to make the context clear.

- Please provide a better resolution for figures from Figure 2 to Figure 6.

-

- Table 10 is difficult to follow in the current format, please consider a different way of organization for better display of information.

- Please highlight in more details the most important descriptors that were by most of the techniques and define their chemical effect briefly when possible.

Reviewer #3: Authors presented a well written thesis on the best practice of machine learning for toxicity prediction.

- 1. Introduction, pp 7-8, lines 103-120: This paragraph currently seemed to be floating out of context, it is recommended to move the paragraph to Line 77 after the first paragraph.

- 3. Methods, 2.1 Data curation: It is advised that the author should publicly share both the original dataset and the curated dataset used in the study along with the Python code used for ML model building as to abide by the PLOS journal guidelines (see Data Availability https://journals.plos.org/plosone/s/data-availability and Sharing Code https://journals.plos.org/plosone/s/materials-software-and-code-sharing) in order to facilitate research reproducibility as well as benefitting the scientific community. The author did a good job of highlighting this importance in section 3.6.1 Reproducibility and it would be great if this manuscript could publicly share the data and code. In regards to the dataset, this would possibly include the compound SMILES, its corresponding bioactivity value and reference source.

- The ML models reported in the manuscript was well carried out and explained in a structured and logical manner.

- The presented guideline is extremely helpful for practitioners in future QSAR model building.

- It is recommended that the authors cite a related paper for the point on research reproducibility https://jcheminf.biomedcentral.com/articles/10.1186/s13321-020-0408-x

- 3. Results and Discussion, 3.6.2 Interpretability, Line 645: Should the term "casually" in the phrase "casually related towards the outputs" a typo and should it be "causally"?

- - 3. Results and Discussion, 3.6.2 Interpretability: Could the authors provide recommendations on how readers could go about interpreting the important descriptors shown in Figures 4-6.

6. PLOS authors have the option to publish the peer review history of their article (what does this mean?). If published, this will include your full peer review and any attached files.

Reviewer #1: No

Reviewer #2: No

Reviewer #3: **Yes: **Chanin Nantasenamat

---

## [Author Response · Author response to Decision Letter 0]

31 Jan 2023

Our responses reviewer comments may be found within the appropriately labelled Word document.

---

## [Decision Letter · Decision Letter 1]

27 Feb 2023

Guidance for good practice in the application of machine learning in development of toxicological quantitative structure-activity relationships (QSARs)

PONE-D-22-28105R1

Dear Dr. Firman,

We’re pleased to inform you that your manuscript has been judged scientifically suitable for publication and will be formally accepted for publication once it meets all outstanding technical requirements.

Kind regards,

Hilal Tayara

Academic Editor

PLOS ONE

Reviewers' comments:

Reviewer's Responses to Questions

**Comments to the Author**

1. If the authors have adequately addressed your comments raised in a previous round of review and you feel that this manuscript is now acceptable for publication, you may indicate that here to bypass the “Comments to the Author” section, enter your conflict of interest statement in the “Confidential to Editor” section, and submit your "Accept" recommendation.

Reviewer #1: All comments have been addressed

Reviewer #3: All comments have been addressed

2. Is the manuscript technically sound, and do the data support the conclusions?

Reviewer #1: Yes

Reviewer #3: Yes

3. Has the statistical analysis been performed appropriately and rigorously? 

Reviewer #1: I Don't Know

Reviewer #3: Yes

4. Have the authors made all data underlying the findings in their manuscript fully available?

Reviewer #1: Yes

Reviewer #3: Yes

5. Is the manuscript presented in an intelligible fashion and written in standard English?

Reviewer #1: Yes

Reviewer #3: Yes

6. Review Comments to the Author

Reviewer #1: For me now the factual scope of the paper is localized to exploring what six commonly used machine learning methods and software can do in toxicology along with 2 popular methods of feature importance. I think that it is done well. I also think that this localization should be emphasized for the following reasons.

For me the selection of the listed methods is not obvious beyond the fact that they are popular and are widely available in the open-source software. All six machine learning methods picked up are considered as not interpretable, requiring a significant effort to extract most important attributes and to conduct other analyses of model interpretability. The used methods of finding important attributes themselves are not universally accepted and criticized in the machine learning literature.

The authors excluded an interpretable single decision tree method as not providing enough accuracy for the studied data. However, there are less popular interpretable methods of sets of logical rules in propositional and first order logic, which are more powerful than individual DTs. These methods are not explored and mentioned. I believe that a wider view of capabilities of such machine learning methods and domain needs to be presented in the paper.

Reviewer #3: All comments and suggestions raised have been addressed by the authors in this revision. It is therefore ready for publication.

7. PLOS authors have the option to publish the peer review history of their article (what does this mean?). If published, this will include your full peer review and any attached files.

Reviewer #1: No

Reviewer #3: No

---

## [Editor Report · Acceptance letter]

27 Apr 2023

PONE-D-22-28105R1 

Guidance for good practice in the application of machine learning in development of toxicological quantitative structure-activity relationships (QSARs) 

Dear Dr. Firman:

I'm pleased to inform you that your manuscript has been deemed suitable for publication in PLOS ONE. Congratulations! Your manuscript is now with our production department. 

Kind regards, 

on behalf of

Dr. Hilal Tayara 

Academic Editor

PLOS ONE